# Training biologically plausible recurrent neural networks on cognitive tasks with long-term dependencies

**Wayne W.M. Soo**
Department of Engineering
University of Cambridge
wmws2@cam.ac.uk

**Vishwa Goudar**
Center for Neural Science
New York University
vishwa.goudar@nyu.edu

**Xiao-Jing Wang**
Center for Neural Science
New York University
xjwang@nyu.edu

## Abstract

Training recurrent neural networks (RNNs) has become a go-to approach for generating and evaluating mechanistic neural hypotheses for cognition. The ease and efficiency of training RNNs with backpropagation through time and the availability of robustly supported deep learning libraries has made RNN modeling more approachable and accessible to neuroscience. Yet, a major technical hindrance remains. Cognitive processes such as working memory and decision making involve neural population dynamics over a long period of time within a behavioral trial and across trials. It is difficult to train RNNs to accomplish tasks where neural representations and dynamics have long temporal dependencies without gating mechanisms such as LSTMs or GRUs which currently lack experimental support and prohibit direct comparison between RNNs and biological neural circuits. We tackled this problem based on the idea of specialized skip-connections through time to support the emergence of task-relevant dynamics, and subsequently reinstitute biological plausibility by reverting to the original architecture. We show that this approach enables RNNs to successfully learn cognitive tasks that prove impractical if not impossible to learn using conventional methods. Over numerous tasks considered here, we achieve less training steps and shorter wall-clock times, particularly in tasks that require learning long-term dependencies via temporal integration over long timescales or maintaining a memory of past events in hidden-states. Our methods expand the range of experimental tasks that biologically plausible RNN models can learn, thereby supporting the development of theory for the emergent neural mechanisms of computations involving long-term dependencies.

## 1 Introduction

The adoption of deep learning methods in neuroscience research has fostered entirely new approaches to modeling neural systems and developing scientific hypotheses [1–4]. Moreover, rapid advances in network initialization and optimization techniques has reduced the threshold knowledge for designing and training neural networks and improved the accessibility of model development in neuroscience. Consequently, it is now far more feasible to formulate theories and even take a theory-first approach to neuroscience. Technological advances in experimental neuroscience are also making it easier to study increasingly complex neural computations, escalating the demand for models that support much longer time-scale dependencies than are conventionally modeled with biologically plausible RNNs (see Appendix A). However, fundamental physiological and anatomical properties of the brain constrain the neural network architectures that are permissible for study in neuroscience. First, the brain's connectivity is highly recurrent at multiple spatial scales, making recurrent neural networks (RNNs) [5] the modeling tool of choice in neuroscience [2, 4, 6]. The training and analysis of RNNs

37th Conference on Neural Information Processing Systems (NeurIPS 2023).

has advanced theory on the neural basis of perception [7–9], motor behavior [10–13], cognition [14–27], memory and navigation [28, 29]. Second, physiological properties of neurons places a greater demand on the emergent computational properties of neural populations. Specifically, in the absence of inputs, a neuron's response rapidly decays away due to its passive membrane properties. Consequently, "leaky" RNN models rely on the reverberation of activity throughout a population of units to integrate inputs over time and to maintain this information in the absence of ongoing inputs. This has even contributed to the population doctrine gaining traction in neuroscience – that neural populations rather than single neurons form fundamental computation units [30, 31].

## 1.1 Gradient instability in recurrent neural networks

The stability challenges of training biologically-plausible (leaky) RNNs with backpropagation through time are widely understood - the larger the number of time steps the network must compute over, the more likely it is that the gradients become catastrophically large or small which arrests further learning. Long-short-term memory (LSTM) [32] and Gated Recurrent Units (GRU) [33] based RNNs were developed precisely to mitigate this adversity posed by exploding or vanishing gradients, doing so by storing memory in self-sustaining cells rather than leaky population activity. Unfortunately, there is a dearth of evidence in neuroscience in support of the gating mechanisms that help LSTMs and GRUs overcome gradient stability issues. There exist multiple biological mechanisms for gating in the brain [34], but none of them is known to turn on and off the neuronal leak, input and output flows in such unconstrained ways as presumed by LSTM or GRU. Moreover, memory function through emergent population dynamics (rather than via a population of self-sustaining units) are a central hypothesis in neuroscience. An extended review of other aspects of biological implausibility of gating mechanisms can be found in Appendix B, which highlights issues with gating properties, ease of training, network dynamics as well as mechanisms underlying memory storage. Other methods that help RNNs learn long-term dependencies have been proposed in the past that do not involve gating mechanisms. A straightforward way would be to simply initialize the recurrent weight matrix as an identity matrix [35]. When applied to ReLU RNNs, these networks were competitive with LSTMs on speech and language benchmarks. Similarly, RNNs whose recurrent weight matrices are constrained to have absolute eigenvalues of 1 have been proposed [36, 37], at the cost of lower convergence rates [38]. At their cores, both methods alleviate the problem by reducing the base multiplier in which gradients explode or vanish. Auxiliary losses targeting intermediate time steps [39] have also been proposed.

## 1.2 Skip connections through time

It is difficult to justify any architectural modifications to biologically plausible RNNs, especially when it involves artificial elements. The alternative, which we propose, is to introduce surrogate elements which improve convergence rates during training and subsequently remove them after training to reinstitute biological plausibility at test time. The ease of adding (and eventually removing) skip connections makes it an ideal candidate for this objective. We present and compare a set of engineering solutions using specialized skip connections for training biologically plausible RNNs on an array of commonly studied cognitive tasks [16]. The idea of using skip connections to improve gradient-based model training gained mainstream popularity when its implementation in deep convolutional neural networks (CNNs) achieved state-of-the-art performance in image classification [40]. Since then, skip connections have become a staple feature of newer architectures in machine learning, such as transformers [41] and MLP-mixers [42]. Skip connections of this nature are also referred to as residual connections. In this context, skip connections solve the degradation problem, where increasing the depth of the model reduces the maximum attainable training performance by direct backpropagation [40]. A conceptually similar idea of skip connections through time involves adding a direct connection between two non-consecutive states in sequential models, thereby facilitating a shortened path during backpropagation and alleviating gradient stability problems. One of the earliest formulations of this idea was proposed more than two decades ago in the training of NARX networks [43]. The architecture of NARX networks naturally contains the mathematical equivalent of skip connections through time, which were referred to as "jump-ahead connections". Network architectures and investigations based on skip connections through time continue to surface even in recent years [44, 45]. For example, dilated skip connections reduce the computational complexity of the RNN, in contrast to regular skip connections which requires additional computations [45]. LSTMs and GRUs are not mutually exclusive with skip connections [46, 47].

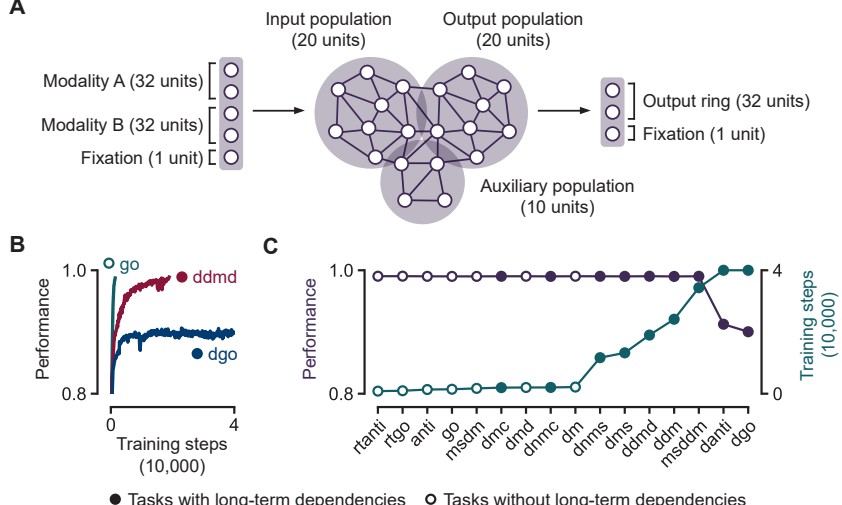

Figure 1: **Limitations of training biologically plausible RNNs for tasks in neuroscience.** **A.** Overview of RNN model. **B.** RNN performance on three cognitive tasks against the number of training steps. RNNs require more training steps in order to successfully perform tasks that require the learning of long-term dependencies. **C.** Number of training steps required to train biologically plausible RNNs to attain 0.99 performance on 16 standard tasks [16] in neuroscience (Appendix D).

## 2 Background

### 2.1 Biologically plausible recurrent neural networks

The continuous-time dynamics of biologically plausible leaky ("vanilla") RNNs are described by:

$$\mathbf{T}\frac{d\mathbf{r}}{dt} = -\mathbf{r} + f\left(\mathbf{W}_{\text{rec}}\mathbf{r} + \mathbf{b} + \mathbf{h}_{\text{ext}} + \boldsymbol{\eta}\right) \tag{1}$$

where $\mathbf{T}$ is a diagonal matrix whose elements are the time constants of each individual neuron (typically constant across all neurons), and $\mathbf{r}$ is the firing rate vector (the hidden-state of the RNN), $\mathbf{W}_{\text{rec}}$ is the recurrent synaptic weight matrix, $\mathbf{b}$ is the bias, $\mathbf{h}_{\text{ext}}$ is the external input vector and $\boldsymbol{\eta}$ represents additive noise present in the system. $f$ is the activation function (or input-output function), which we choose to be ReLU for our simulations. Alternative functions include the hyperbolic tangent (tanh) and softplus. The output of the network is a straightforward readout layer given by

$$\mathbf{y}_t = g(\mathbf{W}_{\text{out}}\mathbf{r}_t) \tag{2}$$

for some activation function $g$, which we choose to be the sigmoid function. Other common choices include softmax, softplus or linear, depending on the decoding objective. These models reflect the diversity in function across the brain areas they model, such perceptual functions in early sensory areas such as the visual cortex [9], maintenance of items in working memory, decision-making, context-dependent or rule-dependent responses in higher-order cortical areas such as the prefrontal cortex [14, 16, 19], and motor control of eye saccades or arm movements in motor cortical areas [10–13]. Consequently, the task structure, requisite computations, RNN inputs and outputs vary widely according to the function. We review the values of some of these parameters from previously-published models [9, 11–25, 29, 48–60] in Appendix C and Table S1 in the supplementary.

### 2.2 Difficulty in learning tasks with long-term dependencies

The suite of cognitive tasks used in our benchmark tests have been adapted from [16], which includes straightforward tasks involving responding after stimulus presentation (go, anti, decision making, decision making with distractors, multi-sensory decision making), reaction time-based tasks (react go/anti), as well as more complex tasks that involve storing information in memory (delayed match/non-match to sample/category, delayed go/anti, delayed decision making). Exact

task descriptions and setup are explained in Appendix D. It has been previously shown that RNNs can learn these tasks without inter-neural interactions, where recurrent connections are restricted to only self-coupling of each neuron [61]. To prevent this class of semi-degenerate and biologically unrealistic solutions, we designated distinct populations of neurons to interface with the input and output units respectively, with the rest of the unassigned neurons designated as auxiliary neurons to aid computations (Figure 1A). This ensures that, at a minimum, there must be connections between the input and output populations, leading to the emergence of meaningful solutions found by the trained networks. For all tasks, the primary inputs and outputs have a ring structure of 32 units, with each unit $i$ having a preferred direction $\psi_i$. An input stimulus is characterized by a modality, a magnitude $\gamma$ and a direction $\psi$ such that:

$$h_i = \gamma \cdot 0.8 \, \exp\left[ -\frac{1}{2}\left( \frac{8|\psi - \psi_i|}{\pi} \right)^2 \right] \tag{3}$$

The input consists of two separate rings, representing two different stimulus modalities. The output is a single ring characterized by a direction $\psi$, representing a saccadic response,

$$y_i = 0.8 \, \exp\left[ -\frac{1}{2}\left( \frac{8|\psi - \psi_i|}{\pi} \right)^2 \right] + 0.05 \tag{4}$$

When cued to "fixate" by a binary fixation input unit, the RNN must respond by maintaining the value of its fixation output unit, constrained to lie between 0 and 1, above 0.5. The desired input-output mapping at other times in a trial are task dependent. The loss function to be minimized is the mean-squared error between the network output and the target output. The network is optimized with respect to the input weights $\mathbf{W}_{\text{inp}}$, output weights $\mathbf{W}_{\text{out}}$, recurrent weights $\mathbf{W}_{\text{rec}}$ and network bias $\mathbf{b}$. For each task, we train 100 models in parallel with time steps ranging from 1500 to 2500 depending on the task, time constant of 100 ms and $dt = 5$ ms, which together represents a higher temporal resolution than typical models (see Table S2 for detailed model parameters). We train each set of 100 models until the average performance of the networks in the $10^{th} - 90^{th}$ percentile reaches 0.99 or until 40,000 training steps have been reached (Figure 1B-C). The number of training steps required for the tasks span more than an order of magnitude (Table 1), thus depicting their varying difficulties. Importantly, we find that tasks that involve some form of long-term dependency typically require more training steps and are harder to train in general. In fact, RNNs trained on two tasks that require long-term dependencies were not able to attain the required threshold performance within 40,000 training steps, but instead maintained a low but constant performance at the later stages of training (Figure 1B).

## 3   Methods

The most common way of training biologically plausible RNNs involves simulating its dynamics using Euler's method for the forward pass (Figure 2A):

$$\mathbf{r}_{t+\Delta t} \leftarrow \mathbf{r}_t + \frac{d\mathbf{r}_t}{dt}\Delta t \tag{5}$$

Here, $\Delta t$ is the discretization time step that plays a central role in performing backpropagation through time on the simulated dynamics. Consequently, iterating this over $\theta$ time steps results in:

$$\mathbf{r}^{\text{base}}_{t+\theta\Delta t} \leftarrow \mathbf{r}_t + \sum_{k=0}^{\theta-1} \frac{d\mathbf{r}_{t+k\Delta t}}{dt}\Delta t \tag{6}$$

which we will denote as the dynamics of the base model $\mathbf{r}^{\text{base}}_{t+\theta\Delta t}$. We explore several ways in which this process can be modified, only during training, in order to help with the learning of long-term dependencies.

### 3.1   Coarsened discretization (CD)

CD begins by training the network with a large step size to support stable gradients while learning long-term dependencies (Figure 2B). Here, (5) is modified to:

$$\mathbf{r}^{\text{CD}}_{t+\theta\Delta t} \leftarrow \mathbf{r}_t + \frac{d\mathbf{r}_t}{dt}\Delta t \times \theta \tag{7}$$

for some discretization factor $\theta > 1$. Thereafter, $\theta$ is gradually reduced over the course of training until $\theta = 1$ at the end of training, resulting in a reversion to the base model described by (5).

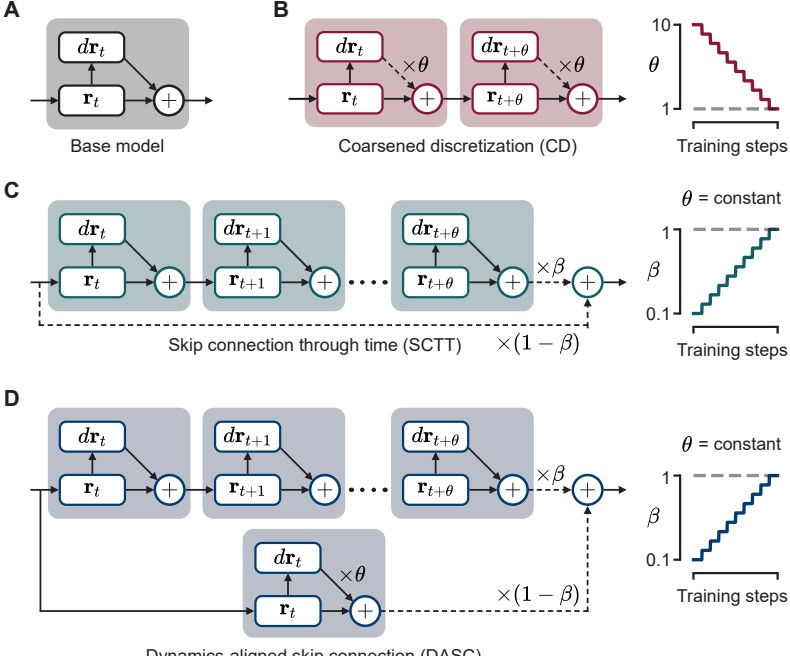

Figure 2: **Proposed algorithms for training RNNs to perform tasks requiring long-term dependencies. A.** Base model: simulating the RNN using Euler's method and training by backpropagation through time. **B.** Coarsened discretization (CD): training in the same way as the base model, except with a larger discretized time step (by a factor of $\theta$) and then gradually reducing $\theta$ as training progresses until $\theta = 1$. **C.** Skip connections through time (SCTT): implementing a skip connection every $\theta$ time steps which takes a weighted average of the hidden state on the two ends of the skip connection with weights $\beta$ and $1 - \beta$, and then gradually increasing $\beta$ until $\beta = 1$. **D.** Dynamics-aligned skip connections (DASC): SCTTs but with a CD step before taking the weighted average.

## 3.2 Skip connections through time (SCTT)

SCTT similarly confines the well-established idea of skip connections through time to training only (Figure 2C). For some mixing ratio $\beta$, a skip connection between time steps $t$ and $t + \theta$ alters the hidden state as:

$$\mathbf{r}^{\text{SCTT}}_{t+\theta\Delta t} \leftarrow (1 - \beta)\, \mathbf{r}_t + \beta\, \mathbf{r}^{\text{base}}_{t+\theta\Delta t} \tag{8}$$

The skip length $\theta$ remains unchanged throughout training, while the mixing ratio $0 < \beta \leq 1$ starts at a small value. This provides shortcuts for gradient backpropagation, thus reducing the risk of gradient instability. Similar to CD, the mixing is gradually reduced (i.e. $\beta$ is gradually increased to 1) until we are left with a trained network without skip connections (5).

## 3.3 Dynamics-aligned skip connections (DASC)

We propose an additional method, DASC, which combines the essence of CD with SCTT, in order to respect dynamical properties of the system (Figure 2D). DASC achieves this by simulating large time steps through its skip connections. The skip connections between $t$ and $t + \theta$ alters the hidden state as:

$$\mathbf{r}^{\text{DASC}}_{t+\theta\Delta t} \leftarrow (1 - \beta)\, \mathbf{r}^{\text{CD}}_{t+\theta\Delta t} + \beta\, \mathbf{r}^{\text{base}}_{t+\theta\Delta t} \tag{9}$$

DASC aims to combine the benefits of CD and SCTT while limiting the risk of gradient instability. Unlike CD, DASC simultaneously trains the model with large and appropriately small time steps; unlike SCTT, it mitigates misalignment in the network's dynamics when it mixes hidden state estimates through the paths with and without skip connections.

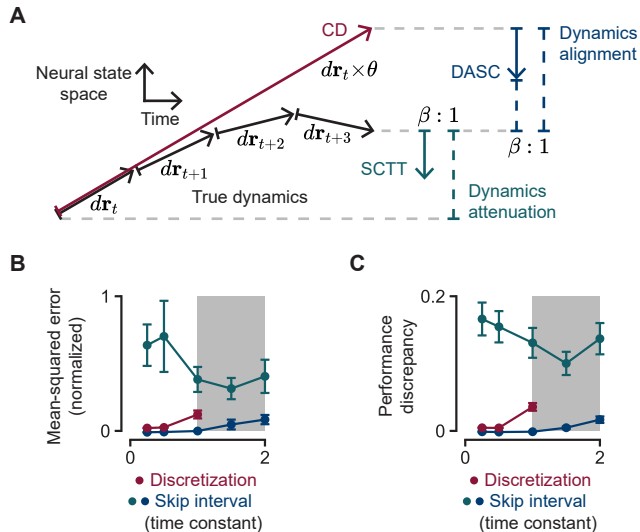

Figure 3: **Effects of proposed algorithms on network dynamics. A.** Diagram of proposed dynamics in activity subspace over time. Coarsened discretization (CD, red) typically results in a large discretization error. Skip connections through time (SCTT, green) result in an attenuation in dynamics. Dynamics-aligned skip connections (DASC, blue) make its hidden-states more aligned with the true dynamics compared to CD. **B.** Normalized mean-squared error between true and modified dynamics for $\beta = 0.5$, $\theta \in \{5, 10, 20, 30, 40\}$. Shaded area $(\theta > 20)$ implies a time discretization greater than the time constant. **C.** Difference in task performance computed between modified and true dynamics.

## 3.4 Network dynamics

We define the neural dynamics of an RNN in the absence of skip connections or time-step over-discretization as its true dynamics; RNNs simulated with any of the methods under study (CD, SCTT or DASC) will thus result in modified dynamics. We provide an intuition of the resultant modified dynamics underlying each method with reference to the true dynamics (Figure 3A). Under reasonable circumstances, CD typically provides a relatively poor estimation of the true network activity as a result of extrapolating over a longer time interval. DASC bridges this discrepancy by taking a weighted average between the true neural activity and neural activity produced by CD (Figure 3A, blue), and is therefore more aligned to the true dynamics than CD by design. We refer to this property as being "dynamics-aligned" (hence its name). SCTT, on the other hand, takes a weighted average between the true network state at two different time points. From the perspective of the later time point, the dynamics can be said to be attenuated (Figure 3A, green).

In order to quantify the effects of modified dynamics on trained networks, we train 100 networks on the aforementioned 16 standard tasks [4] and evaluated the networks with performance in the $10^{th} - 90^{th}$ percentile over 400 training steps using all three proposed methods. For SCTT and DASC, we keep the mixing ratio $\beta$ constant at 0.5, while varying the skip length $\theta \in \{5, 10, 20, 30, 40\}$, where the time constant is 20 times the base discretization step. For CD, we vary the discretization factor, also referred to as $\theta$, using the same 5 values. For all three methods, $\theta$ represents the number of skipped time steps which effectively shortens the minimum number of steps over which backpropagation through time is performed to alleviate gradient stability problems. Therefore, they are plotted on the same axis (Figure 3B-C). We find that it is difficult to train networks with CD using $\theta$ values greater than the time constant. We provide an empirical rationale for this in Appendix E. In contrast, SCTT and DASC remained stable even with skip lengths equal to twice the time constant, which highlights a key advantage in using skip connections. To compare the accuracy of the methods, we compute the mean-squared distance in neural activity between each of their true and modified dynamics for each network and average this distance over all networks and across all tasks (Figure 3B). SCTT's discrepancy is consistently large due to the fact that it offers a zeroth-order approximation of DASC, as shown in the first terms of the RHS of (8) and (9). Conversely, DASC uses a first-order approximation to improve numerical accuracy along the skip connections. These results are mirrored by the discrepancy in output performance, averaged here in the same manner as the mean-squared distance (Figure 3C).

Table 1: **Training biologically plausible RNNs on cognitive tasks requiring long-term dependencies.** For each task, 100 models were simultaneously trained by backpropagation through time until the average performance of the networks in the $10^{th} - 90^{th}$ percentile reaches 0.99 or when 40,000 training steps have been reached. Control represents the base model in Figure 2A. The other columns are methods introduced in Figure 2B-D.

| Tasks that require long-term dependencies | | | | | | |
|---|---|---|---|---|---|---|
| Task | Control | $CD_{10}$ | $SCTT_{10}$ | $SCTT_{40}$ | $DASC_{10}$ | $DASC_{40}$ |
| Training steps ($\times 10^4$) | | | | | | |
| dmc | **0.196** | 0.251 | 1.089 | 0.907 | 0.244 | 0.343 |
| dnmc | **0.203** | 0.282 | 1.089 | 0.916 | 0.252 | 0.558 |
| dnms | 1.167 | 1.311 | 1.269 | 1.299 | **0.993** | 1.065 |
| dms | 1.324 | 1.079 | 1.293 | 1.282 | 1.469 | **1.041** |
| ddmd | 1.900 | 1.465 | 1.270 | **1.018** | 1.757 | 1.059 |
| ddm | 2.413 | 2.030 | 1.283 | 1.088 | 2.129 | **1.055** |
| msddm | 3.434 | 2.096 | 1.269 | **1.071** | 2.196 | 1.121 |
| danti | failed | failed | 1.268 | 1.088 | failed | **1.005** |
| dgo | failed | failed | 1.106 | 1.268 | failed | **1.003** |
| Wall-clock training time ($\times 10^4$s) | | | | | | |
| dmc | 0.085 | **0.016** | 0.191 | 0.156 | 0.119 | 0.167 |
| dnmc | 0.088 | **0.018** | 0.171 | 0.172 | 0.125 | 0.273 |
| dnms | 0.501 | **0.237** | 0.498 | 0.554 | 0.491 | 0.517 |
| dms | 0.565 | **0.105** | 0.515 | 0.499 | 0.723 | 0.508 |
| ddmd | 0.820 | **0.326** | 0.484 | 0.595 | 0.863 | 0.518 |
| ddm | 1.030 | 0.632 | 0.548 | 0.664 | 1.023 | **0.523** |
| msddm | 1.464 | 0.683 | 0.634 | 0.595 | 1.079 | **0.543** |
| danti | failed | failed | **0.402** | 1.345 | failed | 0.409 |
| dgo | failed | failed | 0.325 | **0.261** | failed | 0.412 |
| Floating point operations ($\times 10^{13}$) | | | | | | |
| dmc | 0.114 | **0.080** | 0.229 | 0.187 | 0.144 | 0.201 |
| dnmc | 0.119 | **0.090** | 0.207 | 0.207 | 0.148 | 0.327 |
| dnms | 0.682 | **0.416** | 0.599 | 0.685 | 0.585 | 0.623 |
| dms | 0.773 | **0.343** | 0.625 | 0.609 | 0.865 | 0.609 |
| ddmd | 1.110 | **0.465** | 0.582 | 0.706 | 1.034 | 0.620 |
| ddm | 1.409 | 0.645 | 0.667 | 0.804 | 1.254 | **0.617** |
| msddm | 2.005 | 0.666 | 0.768 | 0.727 | 1.293 | **0.656** |
| danti | failed | failed | **0.575** | 2.062 | failed | 0.588 |
| dgo | failed | failed | 0.465 | **0.386** | failed | 0.587 |

## 4   Results

We train networks on the same set 16 cognitive tasks [16] as in the previous section. For each method, we train 100 separate networks across different hyperparameter values (see Appendix F) until the average performance of the networks in the $10^{th} - 90^{th}$ percentile reaches 0.99, or until a maximum of 40,000 training steps. We then verify that the performance of each network that failed to attain the required performance threshold has converged and remains constant, albeit at a lower level of performance (for example, Figure 1B, blue). The hyperparameters are:

• **Step count.** CD reverts to the base model when $\theta = 1$, while SCTT and DASC revert to the base model when $\beta = 1$. Training starts at some initial value $\theta_0 > 1$ (CD) or $\beta_0 < 1$ (SCTT or DASC). We progressively update the value of $\theta$ or $\beta$ up to a value of 1 (Figure 2, right). The number of steps over which we carry out this progressive update through the course of training is defined as the step count. We consider three values of step counts: $\{1, 5, 10\}$. This annealing process is performed over the first 20,000 training steps in equal intervals.

- **Skip length.** As described in the previous section, all methods have some way of shortening the number of backpropagation steps during training. For SCTT and DASC, this is facilitated by the skip connections. The skip length is therefore defined as the temporal length of the skip connection, which takes three values $\theta \in \{10, 20, 40\}$, representing half, one and twice the time constant respectively. For CD, the skip length is defined as initial discretization factor (i.e. prior to progressive updates), $\theta_0$, which can only take the value of 10 but not 20 and 40 due to network instability problems, as explained in Section 3.4 and Appendix E.

- **Initial ratio.** For SCTT and DASC, we also vary the initial weighting ratio $\beta_0 \in \{0.2, 0.5, 0.8\}$. There is no equivalent term for CD.

As an example, suppose we train using DASC with step size 5 and initial ratio 0.5. Then, $\beta$ would start at 0.5 and then progressively take the values of $0.5, 0.6, 0.7, 0.8, 0.9$ and finally $1.0$ in fixed intervals over the course of training. In summary, we trained 4,800 networks using CD and 43,200 networks each using SCTT and DASC, totalling over 90,000 trained networks (Table S3). For each training session, we recorded the number of training steps (either due to attaining 0.99 performance or hitting the training step limit), the wall-clock time spent in training, and the number of floating point operations involved during training. We highlight the advantage of using skip connections over timescales longer than the time constant by comparing six training configurations: the base model (control), CD with discretization factor 10 ($CD_{10}$), SCTT with skip lengths 10 and 40 ($SCTT_{10}$, $SCTT_{40}$) and DASC with skip lengths 10 and 40 ($DASC_{10}$, $DASC_{40}$) on tasks that require long-term dependencies (9 out of 16 tasks) in Table 1. Detailed results can be found in Appendix F.

In the following analyses, we compare the training performance of methods we introduce here, namely SCTT and DASC, to previously known methods, namely CD and direct training (although no explicit literature was found for CD with the best of our efforts). Skip connections alleviate gradient stability problems during training which improves training efficiency by facilitating a shorter path for backpropagating through time. The most suitable metric to evaluate this objective is the number of training steps needed to learn the tasks (Table 1, top). RNNs trained with SCTT or DASC required fewer training steps than control or CD networks on 7 of the 9 tasks that require long-term dependencies. Moreover, control and CD networks were unable to learn 2 of these 9 tasks. This presents a clear advantage of our proposed methods. However, in general, when evaluating training algorithms, other metrics such as wall-clock training time and total floating point operations are more practical and may be more emphasized (Table 1, bottom). In most tasks, while CD requires more training steps, the wall-clock time and floating point operations required for each step is significantly less (determined by the discretization factor). However, CD networks are not always successful at learning long-term dependency tasks. Also, SCTT and DASC achieved shorter training times and required fewer floating point operations than contol networks on 7 of the 9 tasks. Taken together, we believe that these considerations tilt the advantage in favor of SCTT and DASC. Note that the results in Table 1 are computed using optimal hyperparameter configurations. Hyperparameter-specific analyses can be found in Figure S3 and Appendix G.

## 4.1 Strategic placement of skip connections

Another feature of skip connections is that they do not need to be regular or periodic, as implemented in the networks explored thus far. Skip connections can be placed anywhere over the task duration, where they may enhance training performance or inadvertently disrupt the true network dynamics. However, poorly placed skip connections can produce dynamical instabilities during training. Conversely, we show that by introducing additional yet strategically-placed skip connections, we can exploit its properties to improve training efficiency (Figure 4D). To best illustrate this, we train biologically plausible RNNs on a rule-based visuomotor association task [62, 63]. This task is characterized by the need to maintain long-term dependencies across trials so that it may maintain the current rule in memory across trials and detect implicit changes in the task rule to update this memory. Therefore, in contrast to the tasks studied thus far where networks are trained on single task trials, for this task networks must be trained on sequences of consecutive trials. Consequently, these tasks pose a higher level of difficulty for training biologically plausible RNNs due to the sheer number of time steps involved.

The network architecture is different compared to the previous models (Figure 4A) in that there is no separation of input and output populations, the input structure is changed, and the output of the

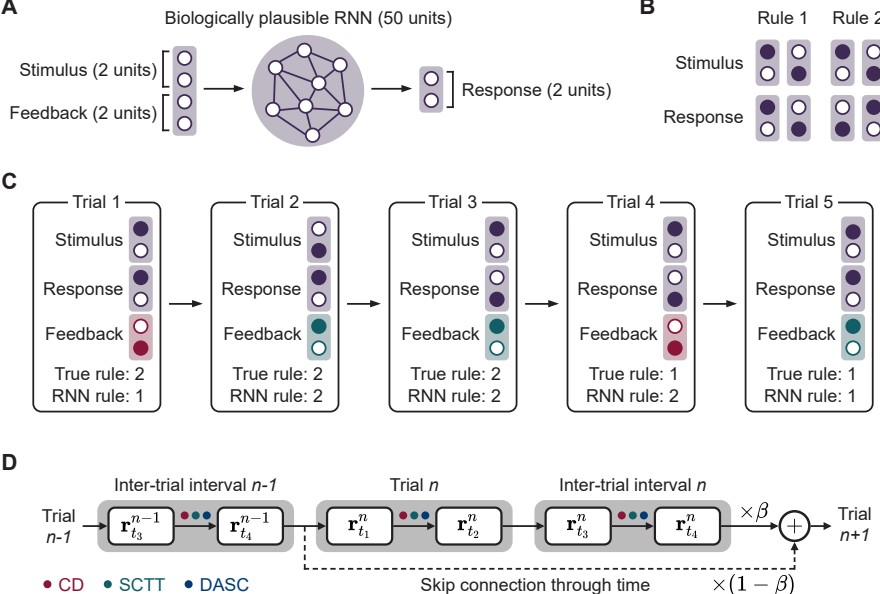

Figure 4: **Strategic placement of skip connections in multi-trial tasks. A.** Overview of RNN model. **B.** Two hidden task rules. **C.** Example of a trial where uncued rule switching occurs. **D.** The task is trained by implementing one of our proposed methods (CD, SCTT, DASC, marked with colored dots) with additional strategically-placed skip connections. In tasks that encompass multiple trials, adding skip connections between the start of every trial provides minimal disruption to network dynamics.

network now passes through a softmax in order to read out a decision:

$$\mathbf{y} = \text{softmax}(\mathbf{W}_{\text{out}}\mathbf{r}) \tag{10}$$

The network receives a 2-dimensional one-hot vector representing one of 2 stimuli as inputs. The target response is dictated by one of two hidden rules. Under Rule 1, the network must associate stimulus 1 with response 1 and stimulus 2 with response 2, and under rule 2 these associations are reversed (Figure 4B). The network may infer the current rule based on the feedback it receives in the form another 2-dimensional one-hot vector that indicates positive or negative feedback following the network's response. Each rule persists across multiple trials, and then switches to the other rule in an uncued manner. Similar to the monkeys in [62], our trained models detect and adapt their behavior within a single trial after the rule switch (Figure 4C). We trained RNNs on 5 consecutive trials, which necessitates learning dependencies that are 5x longer than the previous tasks. In light of this challenge, we define a new metric for performance: the number of networks out of 100 that successfully learn to perform the task at more than 0.99 accuracy. By introducing strategic skip connections between the start of consecutive trials, the proportion of trained networks increased across all methods (Table 2), thereby demonstrating the effectiveness and flexibility of skip connections.

Table 2: **Proportion of networks successfully trained to perform a rule reversal task.** 100 models were trained independently with their respective best hyperparameter configuration for each of the proposed methods and the proportion of successfully trained models are presented as baseline results. This is then repeated with additional strategically-placed skip connections (Figure 4).

| | Proportion of trained networks | | | |
|---|---|---|---|---|
| | Control | $\text{CD}_{10}$ | $\text{SCTT}_{40}$ | $\text{DASC}_{40}$ |
| Baseline | 0.67 ($\pm$0.02) | 0.19 ($\pm$0.02) | 0.71 ($\pm$0.02) | 0.71 ($\pm$0.02) |
| + Strategic SC | **0.76 ($\pm$ 0.02)** | **0.25 ($\pm$ 0.02)** | **0.81 ($\pm$ 0.02)** | **0.78 ($\pm$ 0.02)** |

# 5    Discussion

It can be challenging to train biologically plausible RNNs on cognitive tasks requiring long-term dependencies without the aid of artificial gating elements. In a "vanilla" RNN, an intuitive way to alleviate this difficulty is to coarsen the time discretization (CD) in order to reduce the number of time steps of the overall simulation, thereby shortening the path for backpropagation. However, we show that over-discretization can produce poor approximations of the true continuous-time dynamics (red curves in Figure 3B,C), thus hindering model reliability. While our implementation of CD overcomes this by progressively refining the discretization, our results suggest that it does not offer an improvement in training efficiency (Table 1), even if some time savings is achieved. We also find that its benefits are constrained by the single unit time constant, beyond which the network quickly destabilizes during training (Figure 3B). Finally even with CD, networks may fail to learn challenging long-term dependency tasks (Tables 1, 2). In contrast, SCTT and DASC offer significant improvements in training efficiency and reliability. However, the two approaches accomplish this in different ways. While it provides a shorter path to backpropagate gradients, SCTT inadvertently attenuates the network's dynamics (Figure 3A). This significantly misaligns the network's dynamics in the presence versus absence of the skip connections (Figure 3B,C), which compels more learning as the network must change as the skip connections are annealed away. Instead, DASC emphasizes an alignment of the network's dynamics with (modified) and without (true) skip connections (Figure 3) while still supporting shorter paths for backpropagation.

Experimental advances in neuroscience are facilitating the study of tasks involving increasingly complex and lengthy temporal dependencies. Concurrently, technological advances are enabling simultaneous recordings from increasingly large neural populations. This underscores the need for complementary theory through high-fidelity biologically plausible models that perform challenging computations over lengthy timescales. By introducing methods that enable the development of such models without sacrificing biological plausibility, our work will serve to broaden the applicability of deep learning methods in systems neuroscience. This includes the development of model for evidence accumulation [64], spatial navigation [65], scene processing [66], rule-guided behavior [67, 68] complex rule learning [69], few-shot learning and adaptive behavior [70]. An important reason for using large time steps when training biologically plausible RNNs is that it decreases the computational resources or wall-clock time required for training. While true for simpler tasks, we have shown that there are harder tasks that cannot be learned with conventional methods regardless of time step size.

It is noteworthy that we focus on RNNs built to be directly comparable with neural circuits in the brain, rather than biological plausibility of a training algorithm itself; a neural circuit in the brain is sculpted by not only learning but also development and evolution. In general, training biologically plausible RNNs is a resource-intensive undertaking, and while our methods improve training efficiency and model fidelity, they do not alleviate the resource load per-training step. Additionally, at the present time approaching human-level computational competency requires architectures such as transformers unknown in neuroscience. More work will be necessary to develop biologically plausible yet computationally powerful mechanisms for learning long-term dependencies.

# 6    Conclusion

We have proposed and evaluated a set of methods to learn long-term dependencies in biologically plausible leaky RNNs. These methods improve model fidelity while offering substantial learning performance improvements over conventional training approaches, and even facilitate learning of tasks that are difficult to learn with conventional methods.

## Author Contributions

W.S. developed the algorithm. W.S. and V.G. performed the experiments. All authors designed the study, took part in discussions, interpreted the results, and wrote the paper.

## Acknowledgments and Disclosure of Funding

This work was partly supported by the NIH grant R01MH062349 and ONR grant N00014-23-1-2040.

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
