# Training biologically plausible recurrent neural networks on cognitive tasks with long-term dependencies

**Supplementary material**

## A Demand for biological RNNs with long time-scale dependencies

Biological organisms adapt to changes in their environment in order to survive. Two forms of this ability to adapt that are of particular interest to neuroscience, and that will likely shape future advances in deep learning, is how the brain supports (i) appropriate, sample efficient behavioral changes in response to changes in environmental contingencies (e.g. changing stimuli, reward schedule, or task rules), and (ii) rapid development of solutions to new problems with known and sometimes even unknown (but inferable) structure. Conventional cognitive neuroscience tasks involve computing functions of variables that are observable or inferable from information presented to the subject within the last 2-3 seconds, i.e. within an individual trial. In contrast, tasks involving rapid adaptation or learning often require computing, efficiently updating and tracking task variables across several trials (minutes or even hours), i.e. over arbitrarily longer time-scales than timescale of a single trial.

For example, value-based decision making tasks akin to bandit problems are often used to study the neural basis of economic choices. The values of the choices in these tasks can vary over time (slowly or abruptly depending on the task design) and efficient, adaptive responses to these changes requires tracking the value of the options and making decisions contingent upon the current value estimates [1, 2]. Sometimes, the volatility in the value of the options changes over time and must be tracked in order to adapt appropriately [3]. Strategic or social decision making tasks may invoke game-theoretic strategies wherein subjects must assess their opponents' play to maximize their own payoff [4, 5].

Rule-based tasks form another important class that requires adaptive behavior. In these tasks, the outcome of a subject's response on a trial depends on a hidden rule. The rule changes across blocks of consecutive trials in an uncued manner, but the set of possible rules is small, fixed and known to the subject. Well-trained subjects detect rule switches and infer the new rule rapidly by efficiently integrating outcomes over a few exploratory trials [6–9]. An extension to this is the study of rule-based behavior in situations where a rule does not come from a known set and must be discovered. Poor approximations of the true rule governing the task structure may make it difficult to learn, place unnecessary demands on cognitive resources and adversely affecting performance. Yet, human subjects quickly discover complex task structures with little instruction by via effective learning strategies [10–12]. Moreover, well-learned structures promote generalization thereby speeding up subsequent learning [13].

In all these cases, subjects must draw rapid, accurate inferences based on the feedback they receive in order to adapt to environmental changes, thereby maximizing positive outcomes (total reward, payoff, etc.). One theory holds that this may be achieved via neural dynamics, wherein neural populations track changing environmental contingencies and appropriately alter their computations and decision representations to alter behavior [14, 15]. But exactly how a neural population may achieve this is not known, thus revealing an exciting frontier for the study of the neural basis of adaptive behavior. Yet, the absence of effective methods to train biological RNNs to compute over long timescales (i.e. spanning several trials) poses a fundamental challenge. This challenge extends to tasks that do not require integrating information across trials, but involve long trials instead. This includes spatial navigation, evidence accumulation and decision making tasks wherein the subject must move through a physical space to solve a problem, thus increasing the trial duration and, in some cases, the computational demand of the task.

# B  Biological plausibility of gating mechanisms

Gating-based architectures such as long short-term memory (LSTM) and gated recurrent unit (GRU) were introduced to overcome challenges pertaining to training vanilla RNNs with gradient descent. They endow individual network units with adaptive multiplicative gating applied to their inputs, memory computation and outputs (only for LSTMs). This has two important computational benefits. First, the multiplicative gating, particularly in the context of memory computations, strongly curtails gradient stability challenges [16]. Second, they inherently support adaptive memory timescales and computation at the single unit level. These enhancements make them substantially more computationally powerful than vanilla RNNs, thus inviting their adoption in cognitive models [17, 18] and in phenomenological models of neural computation [19]. However, key biological constraints preclude them from being considered biologically plausible.

- **Gating.** Gating of the inputs and outputs of neurons is thought to be mediated by distinct inhibitory interneuron types via their nonlinear interactions at the dendrites and soma, respectively, of excitatory pyramidal neurons [14]. While it is certainly believed that gating may enhance adaptive properties of neural computation, available evidence and existing models do not support the hypothesis that this gating is multiplicative.

- **Training.** A major motivation for LSTM and GRU architectures is that they are easier to train than vanilla RNNs due to their improved gradient stability. In contrast, RNNs endowed with interneurons and biologically plausible gating functions are more difficult to train than vanilla RNNs [20].

- **Dynamics.** A constraint for biologically-plausible RNN models is that they should produce continuous-time dynamics. LSTM and GRU architectures are discrete-time systems by construction. Continuous-time variants of these architectures have been proposed in the neuroscience literature to address precisely this issue [21]. Moreover, LSTM units typically respond with abrupt activity transitions between consecutive time steps [17]. In contrast, neurons exhibit smooth temporal responses.

- **Memory.** LSTM and GRU memory mechanisms seem too powerful relative to known brain mechanisms of short-term memory function. The forget (update) gate of LSTMs (GRUs) endow individual units with arbitrarily small and large memory timescales. In contrast, the neuron membrane time constants are typically quite small and of limited range. While a variety of Glutamatergic signaling mechanisms are believed to enhance neural integration timescales [22], they are still limited. Moreover, receptor expression levels vary drastically by brain region [23], obviating a general-purpose mechanism to support arbitrary timescales at the single neuron level. Continuous-time neural population dynamics offer a complementary mechanisms for arbitrary (and adaptive) timescales in the brain [24]. There is little evidence for LSTM/GRU like single-unit timescales in the brain.

# C  Review of RNN parameter values in literature

Model parameters of RNNs in literature (Figure S1A) are sometimes restricted to undesirable values due to competing constraints from the perspective of biological-plausibility, numerical accuracy and gradient stability. We review the values of some of these parameters from previously-published models. In particular, only continuous-time RNNs trained by backpropagation through time are considered [25–29, 21, 30–47, 20, 48–52] (more details in Table S1). The value of the neuronal or leak time constant is often set to 100 ms, which reflects the slow dynamics of NMDA receptors in the brain; although values below that are also plausible depending on the proportion of NMDA receptors in the brain area being modeled (Figure S1C). For high numerical accuracy, the value of the discretization time step should be small relative to this time constant. The most common values of the ratio between the time step size and the time constant are 0.1 and 0.2 (Figure S1D). However, these values are still quite large and can produce inaccurate approximations of the underlying system. On the other hand, decreasing the step size increases the model's memory footprint and can potentially introduce vanishing or exploding gradients when training an RNN with backpropagation through time. Indeed, despite the differences in their function, the models we have surveyed seldom use a large number of time steps (Figure S1B). Taken together, we see that the technical challenges of training RNNs with backpropagation through time constrain the duration of the task a network can be trained on or hinder the model's numerical accuracy.

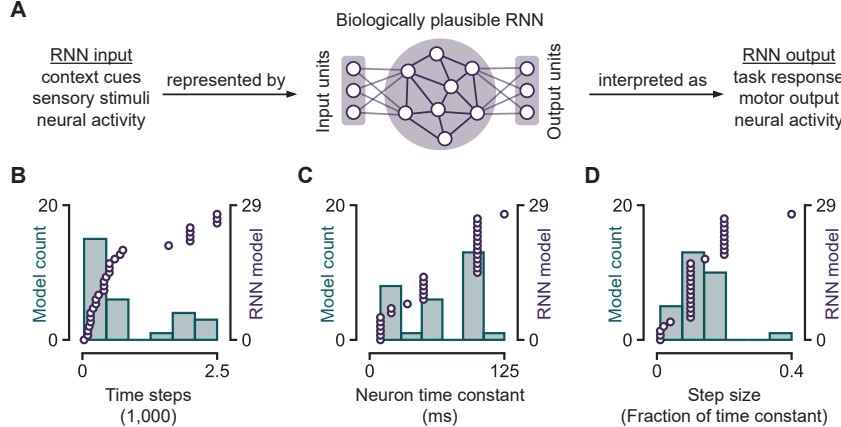

Figure S1: **Biologically plausible RNNs in literature. A.** Overview of how RNN models are used in neuroscience. **B.** Number of time steps used in the respective tasks that the RNNs were trained to perform. **C.** Neuron time constant values in the RNNs. **D.** Step sizes as a fraction of their respective time constants.

## D  Task structures and descriptions

The 16 tasks can be categorized according to their task structures (A, B or C).

**Task structure A.** 7 tasks share this task structure, which consists of 3 task epochs:

$$
\begin{array}{lll}
\text{Task epoch 1} & \text{Fixation} & T_1 \sim \mathcal{U}[T_{\text{short}} - T_{\text{jitter}},\, T_{\text{short}} + T_{\text{jitter}}] \\
\text{Task epoch 2} & \text{Stimulus} & T_2 \sim \mathcal{U}[T_{\text{short}} - T_{\text{jitter}},\, T_{\text{short}} + T_{\text{jitter}}] \\
\text{Task epoch 3} & \text{Response} & T_3 = T_{\text{test}}
\end{array}
$$

- **Go (go):** The fixation cue is set to 1 during the first two epochs. During the stimulus epoch, a stimulus is presented in a randomly chosen modality at a randomly chosen direction. The response, made in task epoch 3, should match the direction of the stimulus.

- **Anti-response (anti):** This is the same as go, but the response should be in the opposite direction of (i.e. $\pi$ radians from) the stimulus.

- **Reaction time go (rtgo):** The fixation cue is set to 1 throughout the trial. During epoch 3, a stimulus is presented in a randomly chosen modality at a randomly chosen direction. The response, made in the same epoch, should match the direction of the stimulus.

- **Reaction time anti-response (rtanti):** Same as rtgo, but the response should be in the opposite direction of (i.e. $\pi$ radians from) the stimulus.

- **Decision making (dm):** The fixation cue is set to 1 during the first two epochs. Two stimuli are simultaneously presented in a single modality during task epoch 2. One stimulus is randomly generated with a random magnitude and direction. The other stimulus is also generated with a random magnitude, but its direction is drawn uniformly at random between $\frac{\pi}{2}$ radians to $\frac{3\pi}{2}$ radians away from the first stimulus. The response, made in task epoch 3, should match the direction of the stimulus with the greater magnitude.

- **Decision making with distractors (dmd):** This is the same as dm, except 2 stimuli are presented in each modality. All 4 stimuli have different magnitudes, but the directions of the two stimuli in modality A are the same as the directions of the two stimuli in modality B. The RNN must ignore modality A and respond in the direction corresponding to the stronger stimulus in modality B.

- **Multi-sensory decision making (msdm):** This is the same as dmd, except the RNN must respond in the direction in which the combined stimulus strength from both modalities is stronger. That is, given stimuli in both modalities at angles $p$ and $q$, if the sum of the magnitudes of the stimuli in both modalities at angle $p$ is larger than the sum of the magnitudes of the stimuli in both modalities at angle $q$, the target response is angle $p$. Otherwise, it is angle $q$.

**Task structure B.** These tasks are similar to tasks with structure A, except they include a delay period between stimulus presentation and response. 2 tasks share this task structure, which consist of 4 task epochs:

| | | |
|---|---|---|
| Task epoch 1 | Fixation | $T_1 \sim \mathcal{U}[T_{\text{short}} - T_{\text{jitter}}, T_{\text{short}} + T_{\text{jitter}}]$ |
| Task epoch 2 | Stimulus | $T_2 \sim \mathcal{U}[T_{\text{short}} - T_{\text{jitter}}, T_{\text{short}} + T_{\text{jitter}}]$ |
| Task epoch 3 | Delay | $T_3 \sim \mathcal{U}[T_{\text{long}} - T_{\text{jitter}}, T_{\text{long}} + T_{\text{jitter}}]$ |
| Task epoch 4 | Response | $T_4 = T_{\text{test}}$ |

- **Delayed go (dgo):** The fixation cue is set to 1 during the first three epochs. A stimulus is presented in a randomly chosen modality with a random direction during task epoch 2. The response made during task epoch 4, i.e. after a delay period, should match the direction of the stimulus.

- **Delayed anti-response (danti):** This is the same as dgo, but the response should be in the opposite direction of (i.e. $\pi$ radians from) the stimulus.

**Task structure C.** These tasks are similar to tasks with structure B, except they include an additional stimulus which is presented after the delay period. The target response is based on the relationship between the two stimuli that are presented before and after the delay period. 7 tasks share this task structure, which consists of 5 task epochs:

| | | |
|---|---|---|
| Task epoch 1 | Fixation | $T_1 \sim \mathcal{U}[T_{\text{short}} - T_{\text{jitter}}, T_{\text{short}} + T_{\text{jitter}}]$ |
| Task epoch 2 | Stimulus 1 | $T_2 \sim \mathcal{U}[T_{\text{short}} - T_{\text{jitter}}, T_{\text{short}} + T_{\text{jitter}}]$ |
| Task epoch 3 | Delay | $T_3 \sim \mathcal{U}[T_{\text{long}} - T_{\text{jitter}}, T_{\text{long}} + T_{\text{jitter}}]$ |
| Task epoch 4 | Stimulus 2 | $T_4 \sim \mathcal{U}[T_{\text{short}} - T_{\text{jitter}}, T_{\text{short}} + T_{\text{jitter}}]$ |
| Task epoch 5 | Response | $T_5 = T_{\text{test}}$ |

- **Delayed decision making (ddm):** The fixation cue is set to 1 during the first four epochs. A stimulus is presented during task epoch 2 in a randomly chosen modality with a random magnitude and direction. After the delay period (task epoch 3), a second stimulus is generated during task epoch 4 in the same modality as task epoch 1 and with a random magnitude, but in a direction that is drawn uniformly at random between $\frac{\pi}{2}$ radians to $\frac{3\pi}{2}$ radians away from the first stimulus. The response, made during task epoch 5, should match the direction of the stimulus with the greater magnitude.

- **Delayed decision making with distractors (ddmd):** This is the same as ddm, except 2 stimuli are presented in each modality. All 4 stimuli have different magnitudes, but the directions of the two stimuli in modality A are the same as the directions of the two stimuli in modality B. The RNN must ignore modality A and respond in the direction corresponding to the stronger stimulus in modality B.

- **Multi-sensory delayed decision making (msddm):** This is the same as ddmd, except the RNN must respond in the direction in which the combined stimulus strength from both modalities is stronger.

- **Delayed match-to-sample (dms):** The fixation cue is set to 1 during the first four epochs. During task epoch 2, a stimulus is presented in a randomly chosen modality with a fixed magnitude and a random direction. After a delay period (task epoch 3), a second stimulus is presented in a randomly chosen modality during task epoch 4 with the same fixed magnitude. In half the trials, the direction of this second stimulus is within $\pm\frac{\pi}{18}$ radians of the first stimulus. This corresponds to a match. In the other half of the trials, the direction of the second stimulus is drawn uniformly between $\frac{\pi}{2}$ radians to $\frac{3\pi}{2}$ radians away from the first stimulus. This corresponds to a non-match. On match trials, the RNN must respond during task epoch 5 in the direction of the second stimulus. On non-math trials, it must continue maintaining fixation during task epoch 5.

- **Delayed non-match-to-sample (dnms):** This is the same as dms, except the RNN must maintain fixation during task epoch 5 on match trials, and respond in the direction of the second stimulus on non-match trials.

- **Delayed match-to-category (dmc):** The fixation cue is set to 1 during the first four epochs. During task epoch 2, a stimulus is presented in a randomly chosen modality with a fixed magnitude. In half the trials, the direction of the first stimulus is drawn uniformly between 0 to $\frac{\pi}{2}$ radians. In the

other half of the trials, the direction is drawn uniformly between $\pi$ radians to $\frac{3\pi}{2}$ radians. After a delay period (task epoch 3), a second stimulus is presented in a randomly chosen modality during task epoch 4 with the same fixed magnitude. The direction of the second stimulus is either in the same quadrant as the first stimulus, corresponding to a match, or in the opposite quadrant which corresponds to a non-match. During task epoch 5 of match trials, the RNN must respond in the direction of the second stimulus. Instead they must continue maintaining fixation during task epoch 5 of non-match trials.

- **Delayed non-match-to-category (dnmc):** This is the same as dmc, except the RNN must maintain fixation during task epoch 5 on match trials, and respond in the direction of the second stimulus on non-match trials.

**Two-choice rule reversal task.** The trial structure consists of 4 task epochs:

| | | |
|---|---|---|
| Task epoch 1 | Stimulus | $T_1 \sim \mathcal{U}[T_{\text{short}} - T_{\text{jitter}}, T_{\text{short}} + T_{\text{jitter}}]$ |
| Task epoch 2 | Response | $T_2 \sim \mathcal{U}[T_{\text{short}} - T_{\text{jitter}}, T_{\text{short}} + T_{\text{jitter}}]$ |
| Task epoch 3 | Feedback | $T_3 \sim \mathcal{U}[T_{\text{short}} - T_{\text{jitter}}, T_{\text{short}} + T_{\text{jitter}}]$ |
| Task epoch 4 | Inter-trial Interval | $T_4 \sim \mathcal{U}[T_{\text{short}} - T_{\text{jitter}}, T_{\text{short}} + T_{\text{jitter}}]$ |

# E  Dynamical stability of proposed methods

We assume that the true dynamics of the network is stable. That is, the network is a stable system when simulated using Euler's method at base time discretization $\Delta t$ with neuron time constant $\alpha \Delta t$. In our simulations, $\alpha = 20$. We observe empirically that when we increase the discretization step size for CD, the network diverges further from its true dynamics and tends to go unstable in general. Particularly, we note that for our networks, the threshold for instability is approximately within a small range around the neuron time constant. To model this, we first consider a linear approximation of the network dynamics:

$$\mathbf{T}\frac{d\mathbf{r}_t}{dt} = -\mathbf{r}_t + f\left(\mathbf{W}\mathbf{r}_t + \mathbf{b} + \mathbf{h}_{\text{ext}} + \boldsymbol{\eta}\right) \tag{1}$$

$$\approx -\mathbf{W}_\theta^* \mathbf{r}_t \tag{2}$$

Here, we assume (from the above empirical explanation) that the dominant eigenvalue of $\mathbf{W}_\theta^*$ is dependent on $\theta$, where $\theta \Delta t$ is the time discretization used to simulate the network. From this framework, this eigenvalue will exceed 1 when $\theta$ reaches some threshold $\theta_{\text{thres}}$. We also define $\mathbf{W}_1^*$ for the case when $\theta = 1$, which is a stable system by definition. Simulating this linearized network for $\theta$ timesteps yields:

$$\mathbf{r}_{t+\theta\Delta t}^{\text{base}} \approx e^{-\theta\frac{1}{\alpha\Delta t}\mathbf{W}_1^*\Delta t}\mathbf{r}_t \tag{3}$$

$$\approx \mathbf{r}_t - \frac{\theta}{\alpha}\mathbf{W}_1^*\mathbf{r}_t \tag{4}$$

Applying the same approximation to the update equation for CD, we find a similar expression:

$$\mathbf{r}_{t+\theta\Delta t}^{\text{CD}} \approx e^{-\frac{1}{\alpha\Delta t}\mathbf{W}_\theta^*\theta\Delta t}\mathbf{r}_t \tag{5}$$

$$\approx \mathbf{r}_t - \frac{\theta}{\alpha}\mathbf{W}_\theta^*\mathbf{r}_t \tag{6}$$

We seek to understand why SCTT and DASC are not subject to the same threshold when increasing the temporal length of their skip connections. In fact, our simulations suggest that these skip connections can extend considerably beyond the CD threshold, up to several multiples of the time constant. This is a huge advantage for SCTT and DASC when it comes to facilitating gradient stability over a large number of time steps. To gain an intuition on why this is the case, we apply the same approximation to the update equation for DASC:

$$\mathbf{r}_{t+\theta\Delta t}^{\text{DASC}} \leftarrow (1-\beta)\,\mathbf{r}_{t+\theta\Delta t}^{\text{CD}} + \beta\,\mathbf{r}_{t+\theta\Delta t}^{\text{base}} \tag{7}$$

$$\approx (1-\beta)\left(\mathbf{r}_t - \frac{\theta}{\alpha}\mathbf{W}_\theta^*\mathbf{r}_t\right) + \beta\left(\mathbf{r}_t - \frac{\theta}{\alpha}\mathbf{W}_1^*\mathbf{r}_t\right) \tag{8}$$

$$= \mathbf{r}_t - \frac{\theta}{\alpha}\left[(1-\beta)\mathbf{W}_\theta^* + \beta\mathbf{W}_1^*\right]\mathbf{r}_t \tag{9}$$

If we define $\mathbf{W}^*_{\text{eff}}$ to be the effective linearized weight matrix:

$$\mathbf{W}^*_{\text{eff}} = (1 - \beta)\mathbf{W}^*_\theta + \beta\mathbf{W}^*_1 \tag{10}$$

so that

$$\mathbf{r}^{\text{DASC}}_{t+\theta\Delta t} \approx \mathbf{r}_t - \frac{\theta}{\alpha}\mathbf{W}^*_{\text{eff}}\mathbf{r}_t \tag{11}$$

we find that it is of the same form as the linear solutions described in (3) and (5), implying that the stability of DASC depends on the eigenvalues of $\mathbf{W}^*_{\text{eff}}$. While $\mathbf{W}^*_\theta$ could have eigenvalues that are greater than 1, we know that $\mathbf{W}^*_1$ gives rise to a stable system (eigenvalues less than 1). The suggests that the overall weighted sum of the two weight matrices may not always lead to an unstable system. From this analysis, we conclude that it is therefore possible to implement skip connections that span longer than the limit faced by CD.

## F    Full training results

The number of training steps taken by all methods across all hyperparameter configurations can be found in Figure S2. For CD, the three bars represent training efficiencies at step counts of 1, 5 and 10 respectively (left to right). For SCTT and DASC, the exact hyperparameter configuration of each bar (left to right) in the figure are given by:

| | | | |
|---|---|---|---|
| 1 | $\theta = 10$ | step count $= 1$ | $\beta_0 = 0.2$ |
| 2 | $\theta = 10$ | step count $= 1$ | $\beta_0 = 0.5$ |
| 3 | $\theta = 10$ | step count $= 1$ | $\beta_0 = 0.8$ |
| 4 | $\theta = 10$ | step count $= 5$ | $\beta_0 = 0.2$ |
| 5 | $\theta = 10$ | step count $= 5$ | $\beta_0 = 0.5$ |
| 6 | $\theta = 10$ | step count $= 5$ | $\beta_0 = 0.8$ |
| 7 | $\theta = 10$ | step count $= 10$ | $\beta_0 = 0.2$ |
| 8 | $\theta = 10$ | step count $= 10$ | $\beta_0 = 0.5$ |
| 9 | $\theta = 10$ | step count $= 10$ | $\beta_0 = 0.8$ |
| 10 | $\theta = 20$ | step count $= 1$ | $\beta_0 = 0.2$ |
| 11 | $\theta = 20$ | step count $= 1$ | $\beta_0 = 0.5$ |
| 12 | $\theta = 20$ | step count $= 1$ | $\beta_0 = 0.8$ |
| 13 | $\theta = 20$ | step count $= 5$ | $\beta_0 = 0.2$ |
| 14 | $\theta = 20$ | step count $= 5$ | $\beta_0 = 0.5$ |
| 15 | $\theta = 20$ | step count $= 5$ | $\beta_0 = 0.8$ |
| 16 | $\theta = 20$ | step count $= 10$ | $\beta_0 = 0.2$ |
| 17 | $\theta = 20$ | step count $= 10$ | $\beta_0 = 0.5$ |
| 18 | $\theta = 20$ | step count $= 10$ | $\beta_0 = 0.8$ |
| 19 | $\theta = 40$ | step count $= 1$ | $\beta_0 = 0.2$ |
| 20 | $\theta = 40$ | step count $= 1$ | $\beta_0 = 0.5$ |
| 21 | $\theta = 40$ | step count $= 1$ | $\beta_0 = 0.8$ |
| 22 | $\theta = 40$ | step count $= 5$ | $\beta_0 = 0.2$ |
| 23 | $\theta = 40$ | step count $= 5$ | $\beta_0 = 0.5$ |
| 24 | $\theta = 40$ | step count $= 5$ | $\beta_0 = 0.8$ |
| 25 | $\theta = 40$ | step count $= 10$ | $\beta_0 = 0.2$ |
| 26 | $\theta = 40$ | step count $= 10$ | $\beta_0 = 0.5$ |
| 27 | $\theta = 40$ | step count $= 10$ | $\beta_0 = 0.8$ |

From these results, we identify the optimal set of hyperparameters for each method for further analysis in the main text. They are:

| | | |
|---|---|---|
| $\text{CD}_{10}$ | Step count $= 1$ | |
| $\text{SCTT}_{10}$ | Step count $= 10$, | $\beta_0 = 0.8$ |
| $\text{SCTT}_{40}$ | Step count $= 10$, | $\beta_0 = 0.8$ |
| $\text{DASC}_{10}$ | Step count $= 1$, | $\beta_0 = 0.5$ |
| $\text{DASC}_{40}$ | Step count $= 1$, | $\beta_0 = 0.5$ |

where the subscript represents the value of $\theta$ for each method (to be precise, it represents $\theta_0$ for CD).

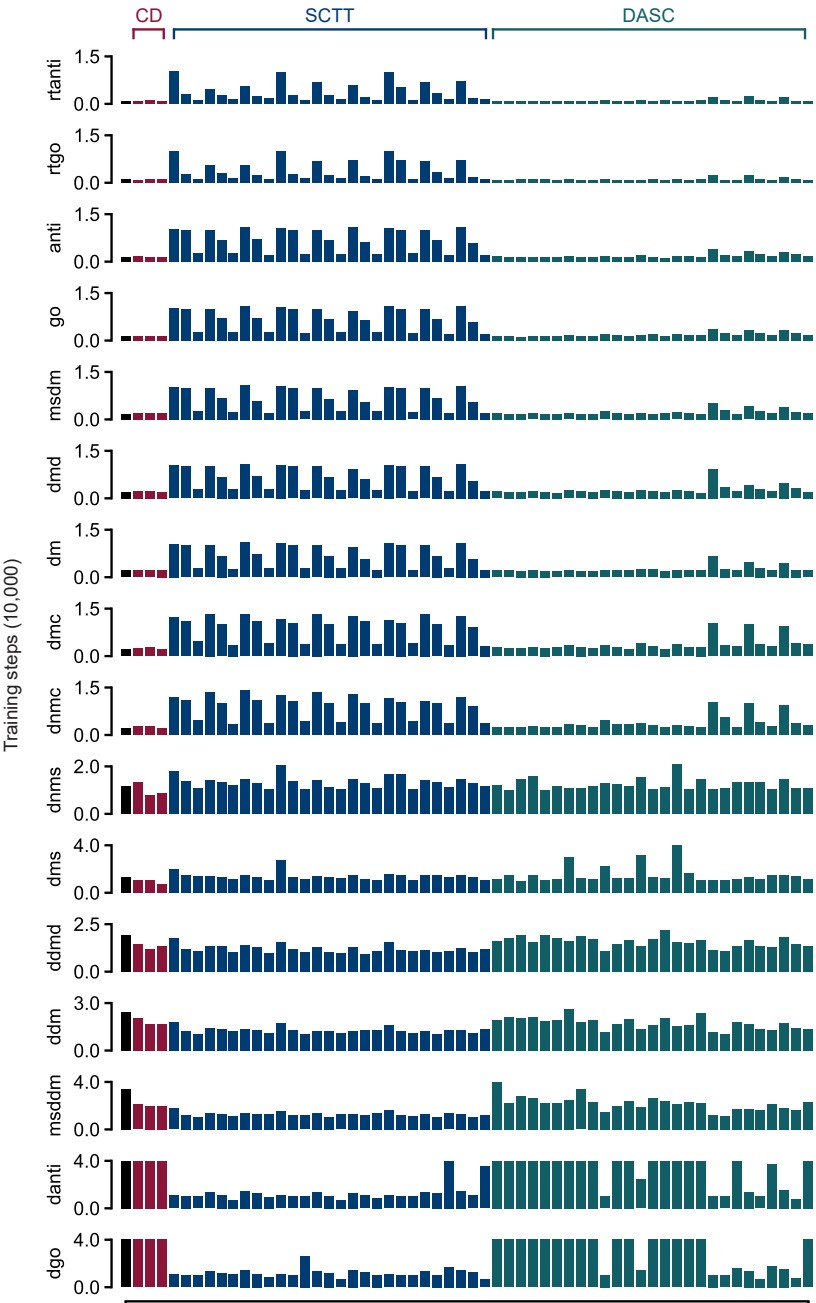

Figure S2: **Training efficiencies of RNNs trained on all 16 standard tasks across all methods in all hyperparameter configurations.** The control model is shown on the left (black). Red, blue and green bars indicate training efficiencies of CD, SCTT and DASC respectively. See section F for exact configurations for each bar.

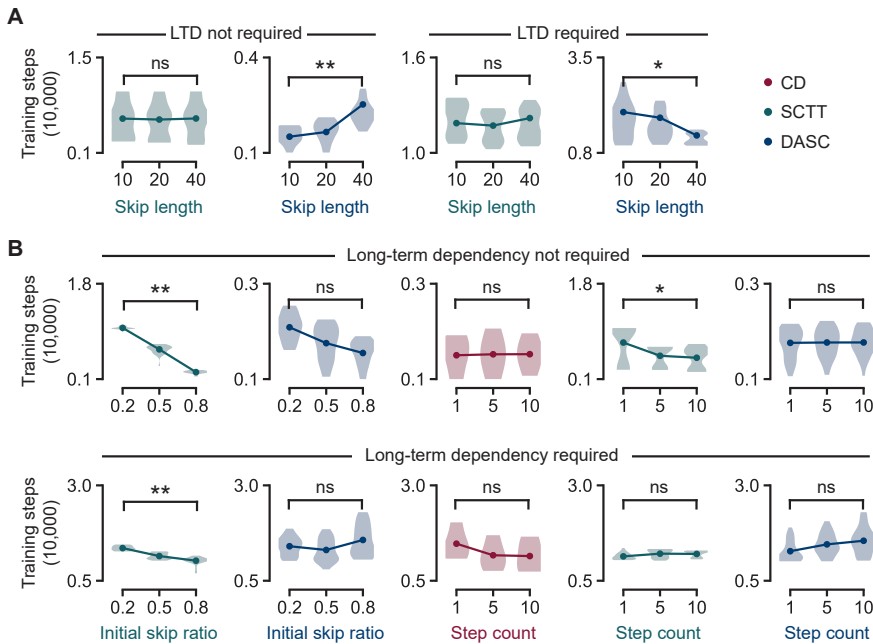

Figure S3: **Analysis of factors affecting training efficiency of each algorithm. A.** Training steps pooled over all 16 tasks against skip length used by SCTT (green) and DASC (blue). A skip length of 20 corresponds to a time interval of one time constant. Plots are differentiated between tasks that require long-term dependencies and tasks that do not. **B.** Effects of the initial skip ratio (left) and step count (right) on training efficiency of each respective method.

Wilcoxon signed-rank test confidence levels: $^*p < 5 \times 10^{-5}$, $^{**}p < 5 \times 10^{-10}$

## G   Hyperparameter exploration

We find that the optimal configuration for DASC strongly depends on the task specifics (Figure S3A). For tasks that do not require maintenance of variables in neural activity-based memory (i.e. over long durations), a short skip length supports modified dynamics that are faithful to the true dynamics, while still providing the advantages of skip connections. Instead, longer skip connections alleviate gradient stability problems that naturally arise in tasks that do require memory, thus improving training efficiency on these tasks. In addition, we find that the training efficiency of models with SCTT depends on the annealing schedule for $\beta$ (Figure S3B). Models train better with SCTT when their training starts at high values of $\beta$ so that their modified dynamics better approximate the true dynamics. They also train better with annealing schedules that change $\beta$ more gradually to avoid abrupt changes in network dynamics during training, thus increasing the number of steps to complete training. This is not observed in CD and DASC, which are not as sensitive to their annealing schedules or to the initial value of $\beta$ (DASC).

## H   Additional training details

All models were trained on high-performance computing clusters consisting of NVIDIA A100 80GB GPUs. Computing resources are summarized in Table S3, showing that we have trained over 90,000 models over more than 1,000 GPU hours. The code to training all models can be found at:

```
https://github.com/wmws2/temporalskip
```

Table S1: **Model parameters of biological RNNs in literature.**

| Ref. | Author(s) (Year) | Time steps | Time constant (ms) | Step size (ms) |
|------|------------------|------------|--------------------|----------------|
| [51] | Chaisangmongkon *et al.* (2017) | 440 | 100 | 10 |
| [39] | Cueva and Wei (2018) | 500 | 10 | 1 |
| [36] | Cueva *et al.* (2020) | 700 | 100 | 10 |
| [31] | Driscoll *et al.* (2022) | 2500 | 100 | 20 |
| [34] | Dubreuil *et al.* (2022) | 125 | 100 | 20 |
| [32] | Echeveste *et al.* (2020) | 2500 | 20 | 0.2 |
| [41] | Goudar *et al.* (2023) | 2000 | 100 | 1 |
| [52] | Kalidindi *et al.* (2021) | 150 | 50 | 10 |
| [47] | Keller *et al.* (2020) | 500 | 10 | 2 |
| [46] | Kim *et al.* (2019) | 200 | 35 | 5 |
| [30] | Kim and Sejnowski (2020) | 500 | 125 | 5 |
| [26] | Kleinman *et al.* (2021) | 150 | 50 | 10 |
| [45] | Liebe *et al.* (2022) | 600 | 100 | 10 |
| [25] | Mante *et al.* (2013) | 750 | 10 | 1 |
| [33] | Masse *et al.* (2019) | 250 | 100 | 10 |
| [35] | Michaels *et al.* (2020) | 300 | 100 | 10 |
| [42] | Murray (2019) | 1600 | 10 | 1 |
| [48] | Orhan and Ma (2019) | 150 | 100 | 10 |
| [44] | Rajakumar *et al.* (2021) | 100 | 50 | 10 |
| [38] | Saxena *et al.* (2022) | 2000 | 10 | 4 |
| [20] | Song *et al.* (2016) | 400 | 50 | 10 |
| [21] | Song *et al.* (2017) | 400 | 100 | 10 |
| [29] | Soo and Lengyel (2022) | 2500 | 20 | 0.2 |
| [40] | Stroud *et al.* (2023) | 2000 | 50 | 1 |
| [27] | Sussillo *et al.* (2015) | 400 | 50 | 5 |
| [49] | Wang *et al.* (2017) | 2000 | 10 | 1 |
| [28] | Yang *et al.* (2019) | 100 | 100 | 20 |
| [43] | Zhang *et al.* (2022) | 30 | 100 | 20 |
| [50] | Zhou *et al.* (2021) | 250 | 100 | 10 |

*Note that if a given reference contains several models, then the parameters of the most relevant model are reported here (if equally relevant, then one is chosen at random).

Table S2: **Model parameters.**

### Task structure

| Variable | Description | Real time | Time steps |
|---|---|---|---|
| $T_{short}$ | Mean time interval of a short task epoch | 500 ms | 100 |
| $T_{long}$ | Mean time interval of a long task epoch | 1000 ms | 200 |
| $T_{jitter}$ | Maximum deviation of mean time intervals | 100 ms | 20 |
| $T_{test}$ | Time interval for test epoch | 500 ms | 100 |
| $T_{wait}$ | Grace period before readout during test epoch | 100 ms | 20 |

### Proposed methods (CD, SCTT, DASC)

| Variable | Description | Real time | Time steps |
|---|---|---|---|
| $(\theta \Delta t)_{max}$ | Maximum step size in CD | 50 ms | 10 |
| $(\beta \Delta t)_{min}$ | Minimum skip size in SCTT & DASC | 50 ms | 10 |
| $(\beta \Delta t)_{max}$ | Maximum skip size in SCTT & DASC | 200 ms | 40 |

### RNNs (general)

| Variable | Description | Initialization | Optimized |
|---|---|---|---|
| $dt$ | Discretized time step | 5 ms | No |
| $\tau$ | Neuron time constant | 100 ms | No |
| $\sigma$ | Standard deviation of input noise | 0.1 | No |
| $\tau_\eta$ | Time constant of input noise | 100 ms | No |
| $\lambda_W$ | Weight regularization coefficient | $10^{-5}$ | No |
| $\lambda_r$ | Activity regularization coefficient | $10^{-5}$ | No |
| $\mathbf{W}_{inp}$ | Input weights | $\mathcal{N}(w_{ij}; 0, 1)$ | Yes |
| $\mathbf{W}_{out}$ | Output weights | $\mathcal{N}(w_{ij}; 0, 1)$ | Yes |
| $\mathbf{W}_{rec}$ | Recurrent weights | $\mathcal{N}(w_{ij}; 0, N^{-1})$ | Yes |
| $\mathbf{b}$ | Bias in recurrent units | $\mathcal{N}(b_i; 0, 1)$ | Yes |

### RNNs trained for standard tasks

| Variable | Description | Initialization | Optimized |
|---|---|---|---|
| $N$ | Number of recurrent units | 50 | No |
| $N_{inp}$ | Number of input units | 65 | No |
| $N_{out}$ | Number of output units | 33 | No |
| $N_{max}$ | Maximum number of training steps | 40,000 | No |

### RNNs trained for rule reversal task

| Variable | Description | Initialization | Optimized |
|---|---|---|---|
| $N$ | Number of recurrent units | 50 | No |
| $N_{inp}$ | Number of input units | 4 | No |
| $N_{out}$ | Number of output units | 2 | No |
| $N_{max}$ | Number of training steps | 10,000 | No |

### Adam optimizer

| Variable | Description | Initialization | Optimized |
|---|---|---|---|
| `lr` | Learning rate | $0.1\, N^{-1}$ | No |
| `beta_1` | First moment exponential decay rate | 0.9 | No |
| `beta_2` | Second moment exponential decay rate | 0.999 | No |
| `epsilon` | Second moment exponential decay rate | $10^{-7}$ | No |

Table S3: **Computing resources.**

**A. Total number of models**

| | | Standard tasks | | |
|---|---|---|---|---|
| Method | Tasks | Configs. per task | Models per config. | Total models |
| Control | 16 | 1 | 100 | 1,600 |
| CD | 16 | 3 | 100 | 4,800 |
| SCTT | 16 | 27 | 100 | 43,200 |
| DASC | 16 | 27 | 100 | 43,200 |

| | | Rule reversal task | | |
|---|---|---|---|---|
| Method | Tasks | Configs. per task | Models per config. | Total models |
| Control | 1 | 2 | 100 | 200 |
| CD | 1 | 2 | 100 | 200 |
| SCTT | 1 | 2 | 100 | 200 |
| DASC | 1 | 2 | 100 | 200 |
| | | | Total models | 93,600 |

**B. Total GPU hours**

| | | Standard tasks | | |
|---|---|---|---|---|
| Method | Tasks | Configs. per task | Avg GPU hrs per config. | Total GPU hours |
| Control | 16 | 1 | 1.47 | 23.5 |
| CD | 16 | 3 | 1.07 | 51.4 |
| SCTT | 16 | 27 | 0.94 | 406 |
| DASC | 16 | 27 | 0.86 | 372 |

| | | Rule reversal task | | |
|---|---|---|---|---|
| Method | Tasks | Configs. per task | Avg GPU hrs per config. | Total GPU hours* |
| Control | 1 | 2 | 3.03 | 60.6 |
| CD | 1 | 2 | 3.50 | 70.0 |
| SCTT | 1 | 2 | 3.51 | 70.2 |
| DASC | 1 | 2 | 3.48 | 69.6 |
| | | | Total GPU hours | approx. 1120 |

*Although there are 100 models per configuration for RNNs trained to perform the rule reversal task, only 10 models could be run in parallel at a time due to memory constraints. As such, GPU hours consumed were increased by a factor of 10 when converting to total GPU hours.