# OpenReview forum: "Training biologically plausible recurrent neural networks on cognitive tasks with long-term dependencies"
_NeurIPS.cc/2023/Conference — NeurIPS 2023 poster_

### Official Review · Reviewer_cfQC · 2023-07-03

**Soundness:** 3 good
**Presentation:** 3 good
**Contribution:** 3 good
**Rating:** 7
**Confidence:** 2

**Summary:**

The paper proposes a method for extracting long-term dependencies in biologically plausible leaky RNNs. The motivation of this work is well-presented and has contribution to the neuroscience field.

**Strengths:**

Strengths:

* The paper demonstrates significant improvements in terms of performance when compared to other baseline methods.

* The paper is well structured and has a smooth and concrete flow.

* I find the incorporation of biological domain knowledge in the algorithm quite interesting and insightful.

**Weaknesses:**

Weaknesses

* In terms of presentation, it would be helpful to list the properties of datasets used in the experimental part.

* Comparison with LSTMs and GRUs-based methods should be a part of the experiments in order to prove the superiority of the suggested approach.

**Questions:**

Please consider the things listed in the “Weaknesses” section.

Also please consider providing information regarding any potential future improvements.

**Limitations:**

To my knowledge and understanding, there is no potential negative societal impact of this work. Other limitations please see “Weaknesses”.

---

> ### Author Rebuttal · Authors · 2023-08-10
>
> We thank the reviewer for the positive comments.
>
> > In terms of presentation, it would be helpful to list the properties of datasets used in the experimental part.
>
> All cognitive tasks used to train our networks are currently explained in Appendix C in the supplementary. We will elaborate further on the neuroscientific basis of each task, as well as provide a summary of the experimental data that have been reported in literature in a revised version if given the opportunity.
>
> > Comparison with LSTMs and GRUs-based methods should be a part of the experiments in order to prove the superiority of the suggested approach.
>
> We agree with the reviewer. As a baseline (and tangential) argument, we first would like to establish the biological implausibility of LSTMs and GRUs. This has been summarized in our reply to reviewer fBiB. The next step would naturally be to compare the capabilities of our proposed methods against gating mechanisms, which will entail future work.
>
> We thank the reviewer again for their time.

---

> > ### Comment · Reviewer_cfQC · 2023-08-19
> >
> > I thank the authors for the response. I will keep my original score.

---

### Official Review · Reviewer_nbN6 · 2023-07-05

**Soundness:** 2 fair
**Presentation:** 2 fair
**Contribution:** 2 fair
**Rating:** 6
**Confidence:** 3

**Summary:**

The authors suggest a method for accelerating the training of continuous time vanilla recurrent neural networks. Specifically, they add temporal skip connections during training, and gradually remove them throughout the training process. For tasks with long term memory requirements, this can provide up to 3 times less training steps. The authors also numerically examine different variants of such skip connections. In particular, they use an interpolation between coarse-grained and fine-grained dynamics.



**Strengths:**

The authors highlight an existing problem with RNNs and offer a novel way of addressing it.

**Weaknesses:**

The main claim – faster training – is not verified. Networks with skip connections might be more computationally expensive. Table S3B gives a hint that there is some saving, but this saving is quite modest. It’s not clear what is the relationship between the algorithms and compute time, and how this scales with parameters such as network size.
It was hard to understand the exact form of the equations, due to the use of “before” and “after” labels. I assume (also from figure 3) that the meaning is that the high-resolution dynamics runs in parallel, and the two versions are combined. If this is correct, then equations 7 and 8 seem incorrect. Equation 7 is the case for CD, not for the discretized equation 1. The last term of equation 8 (“before”) also depends on the recurrent connectivity. A rough approximation would probably be $W^\theta$, because W is applied $\theta$ times to generate this term. If the eigenvalues of W are larger than 1, this will amplify them.
The relation to prior literature is very partial. For instance, skip connections are only described in the supplement – without references in the main text. Other approaches for training RNNs on long dependencies (a few references below as an example) are also not discussed.


Trinh, T., Dai, A., Luong, T., & Le, Q. (2018, July). Learning longer-term dependencies in rnns with auxiliary losses. In International Conference on Machine Learning (pp. 4965-4974). PMLR.Le, Q. V., Jaitly, N., & Hinton, G. E. (2015). A simple way to initialize recurrent networks of rectified linear units. arXiv preprint arXiv:1504.00941.
Arjovsky, M., Shah, A., & Bengio, Y. (2016, June). Unitary evolution recurrent neural networks. In International conference on machine learning (pp. 1120-1128). PMLR.



**Questions:**

The use of the term biological throughout the paper is somewhat misleading. It is true that notations vary between fields and subfields, but still – biological is usually reserved for actual biological networks. Terms such as vanilla or continuous time are perhaps more appropriate. The term biologically plausible could also be used, but that is usually used for adding properties above simple rate networks – separation to E and I neurons, spiking, dendrites, short term plasticity, excitability and so on.
Line 106: alpha used without definition
Figure 1: perhaps use log scale for X axis of panels B,D.
Colors in figures are hard to discern (the specific shades of blue and green used).
If I understand correctly, skip connections usually come with trained weights. In this work, they have the untrained weight beta, to allow their eventual removal. This should probably be stressed.
Comparison of true and modified dynamics (line 170, Fig 3B) – I couldn’t fully understand the setup. Were these networks with the same random seed? Why should neural activity be identical? There are many symmetries, such as permutations, that are expected to arise while training. Was some alignment done between the neural trajectories?
Figure 4. The Y axes are very different in every panel. If possible, this should be avoided. If not (different scales of results), this should be stressed to the reader.
Figure 4B – how do these results look when splitting the tasks?
Figure 4B, last panel. Is the bimodal distribution a result of different tasks?
Figure 4B the X labels are not intuitive.
The first paragraph of the discussion is titled “existing methods” but has no references.


**Limitations:**

yes

---

> ### Author Rebuttal · Authors · 2023-08-10
>
> We thank the reviewer for the extensive review. We hope we can address them within the character limit.
>
> > The main claim is not verified.
>
> Our claim was referring to the reduction in number of training iterations. We understand that this would not be sufficient without data on wall-clock training time and total floating point operations, and hence we have included that in Table R1 in the rebuttal PDF. We strengthen this claim to now be up to a 3.1x reduction in training iterations and 2.7x reduction in training time.
>
> This is not the main claim of this work. From Table 1 (main text) and Table R1 (rebuttal PDF), there are tasks in which an RNN cannot be trained on with conventional methods, but this was made possible using some of our proposed methods. Our strongest result would therefore be that our techniques enable RNNs to successfully learn tasks that are otherwise impossible with conventional methods.
>
> Experiments on how our methods scale with network size will entail future work, but since skip connections involve floating operations that linearly scale with network size, we believe that it will not be as significant as the recurrent dynamics which has quadratic scaling with network size.
>
> > It was hard to understand the equations...
>
> DASC is not strictly two parallel time resolutions since they combine and resplit after the interpolation (Figure 2D).
>
> $$ \mathbf{r}\_{t+\theta\Delta t}^\text{SCTT} \leftarrow (1-\beta) \mathbf{r}\_{t} + \beta \mathbf{r}\_{t+\theta\Delta t} $$
>
> $$ \mathbf{r}\_{t+\theta\Delta t}^\text{DASC} \leftarrow (1-\beta)  \left( \mathbf{r}\_{t} + \frac{d\mathbf{r}\_t}{dt} \Delta{t} \times \theta \right) + \beta \mathbf{r}\_{t+\theta\Delta t} $$
>
> We observe empirically that when we increase the discretization step size for CD, the network becomes unstable. The threshold is approximately the time constant. To model this, we consider a linear approximation of the network dynamics:
> $$
>     \mathbf{T}\frac{d\mathbf{r}\_t}{dt} \approx - \mathbf{W}\_\theta^* \mathbf{r}\_t
> $$
>
> The dominant eigenvalue of $\mathbf{W}\_\theta^*$ is dependent on $\theta$, where $\theta\Delta t$ is the time discretization used to simulate the network. $\alpha\Delta t$ is the time constant. From this framework, this eigenvalue will exceed 1 when $\theta$ reaches some threshold $\theta\_\text{thres}$. We also define $\mathbf{W}\_1^*$ for the base case.
>
> We ask: are the skip connections in SCTT and DASC subject to the same threshold? Our simulations suggest that skip connections can extend beyond that. To gain an intuition, we apply the same model to DASC, where:
>
> $$
> \mathbf{r}\_{t+\theta\Delta t} \approx e^{-\frac{\theta}{\alpha}\mathbf{W}\_1^{*}}\mathbf{r}\_t
> \approx \mathbf{r}\_t - \frac{\theta}{\alpha} \mathbf{W}\_1^\* \mathbf{r}\_t
> $$
>
> so that the update equation becomes:
>
> $$
> \mathbf{r}\_{t+\theta\Delta t}^\text{DASC} \leftarrow (1-\beta)  \left( \mathbf{r}\_{t}- \frac{\theta}{\alpha} \mathbf{W}\_\theta^\* \mathbf{r}\_t \right) + \beta \left( \mathbf{r}\_t - \frac{\theta}{\alpha}\mathbf{W}\_1^\*\mathbf{r}\_t \right) = \mathbf{r}\_t - \frac{\theta}{\alpha}\left[ (1-\beta) \mathbf{W}\_\theta^\* + \beta\mathbf{W}\_1^\* \right] \mathbf{r}\_t
> $$
>
> This is equivalent to a linear system with $\mathbf{W}^* = (1-\beta) \mathbf{W}\_\theta^\* + \beta\mathbf{W}\_1^\*$ which is not always an unstable system. We will include this in the supplementary.
>
> > The relation to prior literature is very partial...
>
> We have briefly summarized related works on this topic and will add the following review to the introduction, as well as include an extended review in the supplementary:
>
> “Other methods that help RNNs learn long-term dependencies have been proposed in the past that do not involve gating mechanisms. A straightforward way that was found would be to simply initialize the recurrent weight matrix as an identity matrix (Le et al. 2015). When applied to ReLU RNNs, the networks were competitive with LSTMs on speech and language benchmarks. On the same line of thought, RNNs whose recurrent weight matrices are constrained to have absolute eigenvalues of 1 have been proposed (Arjovsky et al. 2016, Henaff et al. 2016), at the cost of a lower convergence rate (Vorontsov et al. 2017). At their cores, both methods alleviate the problem by reducing the base multiplier in which gradients explode or vanish. Alternatively, auxiliary losses targeting intermediate time steps (Trinh et al. 2018) have been proposed, which reduces the exponents rather than the base of the gradient instability problem. Skip connections fall under this category of solutions.”
>
> Le et al. (2015) doi:10.48550/arXiv.1602.06662
> Arjovsky et al. (2016) doi:10.48550/arXiv.1511.06464
> Henaff et al. (2016) doi:10.48550/arXiv.1602.06662
> Vorontsov et al. (2017) doi:10.48550/arXiv.1702.00071
> Trinh et al. (2018) doi:10.48550/arXiv.1803.00144
>
> > The use of the term biological throughout the paper is somewhat misleading.
>
> We agree with the reviewer that the nomenclature should be intuitive rather than be specific to any field or subfield. We will refer to our networks as biologically plausible (instead of biological) in a revised version to reflect the incorporation of realistic neuron time constants and lack of artificial gating mechanisms.
>
> > Comparison of true and modified dynamics...
>
> We have included a flowchart in Figure R2 in the rebuttal PDF, plus an explanation in the general response. Networks have different seeds and no alignment was performed.
>
> > Figure 4...
>
> See Figure R3 in the rebuttal PDF and the accompanying explanation in the general response. The reviewer’s intuition is correct regarding the bimodal distribution, and we have split the tasks according to long-term dependency requirements in Figure R3.
>
> We have also fixed all other minor issues raised by the reviewer, such as color contrast, undefined symbols and issues with axis labels. We hope that we can convince the reviewer about the significance of our work.

---

> > ### Comment · Reviewer_nbN6 · 2023-08-13
> > **More questions**
> >
> > Thanks for the answers and clarificaions.
> >
> > A couple of questions
> >
> > Regarding the equations:
> >
> > Figure 2D shows multiple steps that are later combined with a single step. The equation you wrote in the rebuttal is a single step. Can you please clarify? I might be missing something here, but there seems to be an inconsistency.
> >
> > Literature: I think curriculum learning is similar in spirit to this work, because there is a modification to the training stage.

---

> > > ### Author Response · Authors · 2023-08-13
> > >
> > > We thank the reviewer for the question. This has greatly helped with how we should address the confusion.
> > >
> > > In a regular simulation with base time discretization $\Delta t$, one would require $\frac{T}{\Delta t}$ steps to simulate some time interval $T$. Performing backpropagation on this simulation leads to vanishing/exploding gradients on many cognitive tasks.
> > >
> > > When increasing the timestep discretization by a factor $\theta > 1$ so that it becomes $\theta\Delta t$, the number of timesteps to simulate this interval would be $\frac{T}{\theta \Delta t} < \frac{T}{\Delta t} $. By reducing the number of time steps, the gradients become more stable, but the dynamics that we simulate is no longer entirely accurate.
> > >
> > > In order to alleviate the gradient stability problem and still have realistically accurate dynamics, we perform the following steps:
> > > 1. Start at some state $\mathbf{r}_t$ at time $t$.
> > > 2. Run the network at $\Delta t$ for $\theta$ steps. This gives us $\mathbf{r}_{t + \theta \Delta t}$. This is one term but comprises of multiple steps. (Top row of Figure 2D)
> > > 3. Run the network at $\theta \Delta t$ for one step. This gives us $\mathbf{r}_{t} + \frac{d\mathbf{r}_t}{dt} \theta\Delta t$.  (Bottom row of Figure 2D)
> > > 4. Linearly combine them with some ratio $\beta$ to give $\mathbf{r}_{t + \theta\Delta t}^\text{DASC}$.
> > > 5. This $\mathbf{r}\_{t + \theta\Delta t}^\text{DASC}$ would become $\mathbf{r}_{t'}$ at time $t'$ where we are back at step 1.
> > >
> > > If the reviewer is confused what happens in-between, say  $\theta' < \theta$ steps into the simulation, we compute them as $\mathbf{r}_{t + \theta' \Delta t}$ without any adjustments from the coarse-grained simulation.
> > >
> > >
> > >
> > > We agree with the reviewer's reference to curriculum learning, where modifications are confined to training only. In formulating our algorithm, we drew inspiration from many such methods, such as GANs (where only one half of the network is eventually used while the other half is only for training), surrogate gradient training in spiking neural networks (where gradients are computed in a different way during training), as well as curriculum learning (where the initial training objective differs from the final training objective). These methods all share the same high-level idea, but all target different problems. In our case, we are focusing on the gradient instabilities during backpropagation through time.
> > >
> > > We thank the reviewer for the kind comments.

---

> > > > ### Comment · Reviewer_nbN6 · 2023-08-13
> > > > **update**
> > > >
> > > > Thanks for the quick answer.
> > > > I understood that this is the point of the DASC. My question in the initial review and now was related to step 2 of your description. Gradient descent still has to go through several iterations to compute the gradient of $r_{t+\theta dt}$.
> > > >
> > > > In any case, the empirical results show that this technique allows you to train tasks for longer intervals than a naive backdrop allows.
> > > > I'm updating my score based on the rebuttal.

---

### Official Review · Reviewer_fBiB · 2023-07-07

**Soundness:** 3 good
**Presentation:** 2 fair
**Contribution:** 2 fair
**Rating:** 4
**Confidence:** 3

**Summary:**

In this study, the authors propose a couple of architectural innovations to train biological RNNs in a stable manner on a suite of 16 cognitive tasks, some of which require the integration of long-range temporal dependencies. The authors motivate their work from the lack of cortical evidence for gating mechanisms that are crucial to high-performing RNNs such as GRU and LSTM. Instead the authors focus on incorporating skip connections through time in various implementations to learn long-term dependencies. Over various experiments and numerical simulations of recurrent trajectories, the authors show that their proposed DASC method to apply skip connections through time performs effectively, especially when the task requires learning long-term dependencies.

**Strengths:**

+ RNNs are widely used in encoder-decoder models trained on cognitive tasks or neural data. The proposed work aims to address one of the issues that plagues RNNs - learning long range dependencies. Skip-connections through time aren't actively explored in RNNs for computational neuroscience and I appreciate the authors' effort in studying this mechanism for improving RNN training stability.
+ The figures in the paper are commendably presented and really help the readers to understand the underlying mechanisms and differences between Coarsened Discretization, Skip Connections Through Time and DASC.

**Weaknesses:**

- My main issue with the paper is that I believe it is not well motivated; they mention that using biological RNNs without gates is important but this claim is not backed with any advantage of biological RNNs over ones with gates. As such, I don't think one would consider gating as biologically implausible as this computation has surfaced in various parts of the cortex.
- A related issue is that, the authors have not shown comparisons of their proposed skip-connections models against the established gated recurrent networks like GRU and LSTM. I find this as an important comparison that is missing, as it allows for relatively measuring the contribution of temporal skip-connections against gating mechanisms.
- I believe the writing in the paper could be improved further, there is a lot of information that isn't central to the paper's theme such as Section 2.2 (Review of model parameters) which obstructs the paper's flow and makes it harder to glean the main message.
- Even though the authors mention a variety of usecases for biological RNNs (such as modeling neural responses), they only evaluate their models on a suite of simple cognitive tasks. It would be more holistic if the authors also included observations on whether the alternate usecases of RNNs in neuroscience also benefit from adding dynamic skip connections through time.

**Questions:**

- My suggestion to the authors is to strengthen their stance against gated recurrent networks; if these gated networks are good models of cognitive tasks and neural data, why should they not be favored over biological RNNs? As such, all these networks are trained with BPTT which also does not have strong biological evidence.

**Limitations:**

The authors have stated limitations of the work adequately.

---

> ### Author Rebuttal · Authors · 2023-08-10
>
> > ... they mention that using biological RNNs without gates is important but this claim is not backed with any advantage of biological RNNs over ones with gates...
>
> We thank the reviewer for raising this important issue. LSTM and GRU architectures were introduced to overcome challenges pertaining to training vanilla RNNs with gradient descent. LSTMs and GRUs endow individual network units with adaptive multiplicative gating applied to their inputs, memory computation and outputs (only for LSTMs). This has two important computational benefits. First, the multiplicative gating, particularly in the context of memory computations, strongly curtails gradient stability challenges (Hochreiter and Schmidhuber, 1997). Second, they inherently support adaptive memory timescales and computation at the single unit level. These enhancements make them substantially more computationally powerful than vanilla RNNs, thus inviting their adoption in cognitive models (Storrs and Kriegeskorte, 2019; Feinman and Lake, 2020) and in phenomenological models of neural computation (Pandarinath et al., 2019). However, key biological constraints preclude them from being considered biologically plausible:
>
> **Gating.** Gating of the inputs and outputs of neurons is thought to be mediated by distinct inhibitory interneuron types via their nonlinear interactions at the dendrites and soma, respectively, of excitatory pyramidal neurons (Wang and Yang, 2018). While it is certainly believed that gating may enhance adaptive properties of neural computation, available evidence and existing models do not support the hypothesis that this gating is multiplicative.
>
> **Training.** A major motivation for LSTM and GRU architectures is that they are easier to train than vanilla RNNs due to their improved gradient stability. In contrast, RNNs endowed with interneurons and biologically plausible gating functions are more difficult to train than vanilla RNNs (Song et al., 2016). In future work, we certainly aim to extend DASC to these models in order to improve their practical application in computational neuroscience studies.
>
> **Dynamics.** A constraint for biologically-plausible RNN models is that they should produce continuous-time dynamics. LSTM and GRU architectures are discrete-time systems by construction. Continuous-time variants of these architectures have been proposed in the neuroscience literature to address precisely this issue (Song et al., 2017).  Moreover, LSTM units typically respond with abrupt activity transitions between consecutive time steps (Storrs and Kriegeskorte, 2019). In contrast, neurons exhibit smooth temporal responses.
>
> **Memory.** LSTM and GRU memory mechanisms seem too powerful relative to known brain mechanisms of short-term memory function. The forget (update) gate of LSTMs (GRUs) endow individual units with arbitrarily small and large memory timescales. In contrast, the neuron membrane time constants are typically quite small and of limited range (Tripathy and Gerkin, 2016). While a variety of Glutamatergic signaling mechanisms are believed to enhance neural integration timescales (Reiner and Levitz, 2018), they are still limited. Moreover, receptor expression levels vary drastically by brain region (Froudist-Walsh, et al., 2023), obviating a general-purpose mechanism to support arbitrary timescales at the single neuron level. Continuous-time neural *population* dynamics offer a complementary mechanisms for arbitrary (and adaptive) timescales in the brain (Vogels et al., 2005). There is little evidence for LSTM/GRU like single-unit timescales in the brain.
>
> For these reasons, our study, which focuses on enhancing the ability of biologically-plausible trained RNNs to compute over long timescales, foregoes the analysis of DASC in LSTM and GRU networks. We also intend to include this review in the supplementary, with the main points summarized in the introduction of the main text.
>
> Hochreiter, S., & Schmidhuber, J. (1997).  doi:10.1162/neco.1997.9.8.1735
> Feinman, R., & Lake, B. M. (2020). doi:10.48550/arXiv.2003.08978
> Storrs, K. R., & Kriegeskorte, N. (2019). doi:10.48550/arXiv.1903.01458
> Pandarinath, C. et al. (2018). doi:10.1038/s41592-018-0109-9
> Wang, X. J., & Yang, G. R. (2018). doi:10.1016/j.conb.2018.01.002
> Song, H. F. et al. (2016). doi:10.1371/journal.pcbi.1004792
> Song, H. F. et al. (2017). doi:10.7554/eLife.21492
> Reiner, A., & Levitz, J. (2018). doi:10.1016/j.neuron.2018.05.018
> Froudist-Walsh, S. et al. (2023). doi:10.1038/s41593-023-01351-2
> Vogels, T. P. et al. (2005). doi:10.1146/annurev.neuro.28.061604.135637
>
> > I believe the writing in the paper could be improved further...
>
> We refer the reviewer to our general response which details a full outline overhaul.
>
> > Even though the authors mention a variety of usecases for biological RNNs (such as modeling neural responses), they only evaluate their models on a suite of simple cognitive tasks...
>
> We believe that our suite of cognitive tasks are a representative sample of what is being trained in the field. The reviewer is right that these are relatively simple tasks; this does not necessarily represent a problem in our work but exposes another problem – that such RNNs are currently unable to learn more complex tasks. Our method expands the range of tasks. We have also provided a review in Appendix A on complex tasks in neuroscience that our method can be applied to.
>
> > ... all these networks are trained with BPTT which also does not have strong biological evidence.
>
> All references in Table S1 have used BPTT to train their networks in order to draw comparisons with the brain. Our work is intended to support this research direction. While the ideal goal would be to train with biological learning rules, our work focuses first on finding a working solution after which other solutions can be explored by implementing more biological constraints.
>
> We thank the reviewer and hope we have cleared most doubts.

---

### Official Review · Reviewer_T2Cs · 2023-07-08

**Soundness:** 4 excellent
**Presentation:** 2 fair
**Contribution:** 3 good
**Rating:** 7
**Confidence:** 4

**Summary:**

This paper describes and tests several methods for training a class of “biologically-plausible” recurrent neural networks (RNN), which are discrete approximations of continuous-time leaky RNNs. This type of gate-less RNN model has difficulty learning dependences over long time intervals, so tweaks are applied during training time to help stabilize and propagate gradients through time. Three such tweaks are explored: coarsened discretization (CD), where the RNN time step is simply increased during training; skip connections through time (SCTT); and the new method dynamics-aligned skip connections (DASC), which cleverly combines CD and SCTT. Some theoretical analyses of these methods are provided, and then each method is evaluated on a variety of realistic cognitive tasks. The results show that the new method, DASC, tends to perform best (measured by how long the model takes to reach a performance criterion), although different tasks require different hyperparameters.


**Strengths:**

* This paper is quite comprehensive, including clear mathematical descriptions (and nice figures) of the methods, theoretical analyses of their properties, and thorough experimental evaluation.
* The overall approach—modify the network at training time, then remove the modifications after training—is a clever end-run around issues of biological plausibility. The resulting models are always in the “biologically plausible” class, even though the training procedure that arrived there is not. (And we suspect that training with backpropagation is implausible anyway, so no big change there.) While I’m not sure I’ve seen this approach taken with this type of RNN in the past, the authors might note that similar ideas have been floated for spiking neural networks by using surrogate loss at train time.
* The results are solid and seem useful going forward!

**Weaknesses:**

The science in this paper is strong, so all the weaknesses noted here relate to clarity and presentation.
* I found many points in this paper fairly difficult to understand or under-explained until I read the Appendix. There is quite a bit of information in the Appendix that, by any right, should be in the main paper. (References to earlier skip-connection work, the actual model formulation used, how training and evaluation was performed—it was unclear from the main paper how or when the extra discretization and skip connections were removed when testing models!—and descriptions of the tasks.) I understand that the authors are operating under strict page limits, but should this paper be accepted I strongly urge the authors to consider moving some things that are likely critical information for a reader into the main text.
* While the figures illustrating the model (Figures 2 & 5) and showing experimental results (Figure 4) were fairly clear, those summarizing earlier work (Figure 1) and illustrating the modified model dynamics (Figure 3) were much more opaque. I’d recommend the authors attempt to simplify or clarify these figures.
* One issue I was left somewhat uncertain of was novelty. As mentioned above, skip connections are by no means new, but are presented as a novel technique in the main text of this paper. I am not deeply embedded enough in the literature to know how to place DASC. I would have felt better about this if the authors had spent a bit more time  & ink on describing earlier approaches to these issues and connections to their proposed methods.

**Questions:**

None noted. (See suggestions under "weaknesses" above.)

**Limitations:**

The discussion of limitations is clear and complete.

---

> ### Author Rebuttal · Authors · 2023-08-10
>
> We thank the author for the positive review and helpful comments. We also thank the reviewer for the suggested relevance to surrogate gradient approaches used in training spiking neural networks – we will mention this in a revised version if given the opportunity.
>
> > I found many points in this paper fairly difficult to understand or under-explained until I read the Appendix ... I strongly urge the authors to consider moving some things that are likely critical information for a reader into the main text.
>
> We apologize for including important information in the supplementary. As specific formatting issues are difficult to address in a rebuttal without providing context to the (modified) overall narrative, we refer the reviewer to our general response to all reviewers which details a full outline overhaul.
>
> > While the figures illustrating the model (Figures 2 & 5) and showing experimental results (Figure 4) were fairly clear, those summarizing earlier work (Figure 1) and illustrating the modified model dynamics (Figure 3) were much more opaque.
>
> We apologize for the confusion regarding Figures 1 and 3. For Figure 1, the data in panels B-D are summarized in Table S1 in the supplementary with their exact values listed. Panel E involves simulations that we have run, whose exact datapoints are summarized in the first data column of Table 1 in the main text. We have also included a step-by-step diagram of how the plots in Figure 3B-C are generated in the rebuttal PDF (Figures R1 and R2), along with further explanation in the general response.
>
> > One issue I was left somewhat uncertain of was novelty.
>
> We would like to clarify the novelty of our work in this reply, which we will include in a revised version if given the chance. For each of the methods in the manuscript, we further classify them into two versions: (1) the regular feedforward version and (2) the version which confines the method to training only.
>
> **Coarsened discretization (CD)**
>
> Regular: We do not regard this idea as novel to the paper.
> The concept of using a larger time discretization for simulations is well-established since the early days of running simulations on a computer using Euler’s method.
> Training only: We do not regard this idea as novel to the paper.
> In general, experiments were run with large discrete steps following verification or assumption that using smaller steps does not change the resultant dynamics.
>
> **Skip connection through time (SCTT)**
>
> Regular: We do not regard this idea as novel to the paper.
> We have provided a brief overview of the background of skip connections through time (currently as Appendix B, but will be included in the main text in a revised version).
>
> Training only: We regard this idea as novel to the paper.
> In existing models, skip connections are either inherent in some mathematically-equivalent form or added as a permanent part of the model architecture. Our unique goal of biological-plausibility has resulted in the rare situation where removing features is justified.
>
> **Dynamics-aligned skip connection (DASC)**
>
> Regular: We regard this idea as novel to this paper.
> Training only: We regard this idea as novel to this paper.
>
> In addition, we also regard the following analyses as original work:
> (1) showing that removing skip connections by gradually reducing their linear weights allows us to successfully bake in long-term dependencies into a vanilla RNN
> (2) analysis of network activity dynamics and stability – the time skip used in CD cannot exceed the time constant of the neuron due to stability issues, while SCTT and DASC can extend skip connections way beyond this interval
> (3) hyperparameter analyses of all three proposed methods and the conclusions drawn, such as using short skip connections for simpler tasks and longer skip connections for tasks requiring long-term dependencies
> (4) strategic placements of skip connections can result in minimal disruption to network activity and improve training efficiency
>
> > ... describing earlier approaches to these issues and connections to their proposed methods.
>
> We present a review of earlier approaches of solving the long-term dependency problem without gating mechanisms, which we plan to add to the introduction, as well as provide a more in-depth review in the supplementary:
>
> “Other methods that help RNNs learn long-term dependencies have been proposed in the past that do not involve gating mechanisms. A straightforward way that was found would be to simply initialize the recurrent weight matrix as an identity matrix (Le et al. 2015). When applied to ReLU RNNs, the networks were competitive with LSTMs on speech and language benchmarks. On the same line of thought, RNNs whose recurrent weight matrices are constrained to have absolute eigenvalues of 1 have been proposed (Arjovsky et al. 2016, Henaff et al. 2016), at the cost of a lower convergence rate (Vorontsov et al. 2017). At their cores, both methods alleviate the problem by reducing the base multiplier in which gradients explode or vanish. Alternatively, auxiliary losses targeting intermediate time steps (Trinh et al. 2018) have been proposed, which reduces the exponents rather than the base of the gradient instability problem. Skip connections fall under this category of solutions.”
>
> Le et al. (2015) doi:10.48550/arXiv.1602.06662
> Arjovsky et al. (2016) doi:10.48550/arXiv.1511.06464
> Henaff et al. (2016) doi:10.48550/arXiv.1602.06662
> Vorontsov et al. (2017) doi:10.48550/arXiv.1702.00071
> Trinh et al. (2018) doi:10.48550/arXiv.1803.00144
>
> Overall, we hope that we are able to clarify any confusion and address the concerns of the reviewer, especially regarding novelty and priority of information in the main text and supplementary.

---

> > ### Comment · Reviewer_T2Cs · 2023-08-14
> >
> > Thank you for these clear and cogent responses. I was pretty happy with the original submission, and the updated information here & in the general comment has solidified that feeling. My only apprehension is that the authors are proposing a fairly major paper reorganization that the reviewers won't be able to read, but I also appreciate how responsive the authors are to our concerns. I will update my overall rating to a 7.

---

### Official Review · Reviewer_CT76 · 2023-08-01

**Soundness:** 3 good
**Presentation:** 3 good
**Contribution:** 2 fair
**Rating:** 5
**Confidence:** 3

**Summary:**

This paper proposes a way to speed up biological RNN training by up to 3x using well placed skip connections during training time.

**Strengths:**

This paper will be of interest to researchers training biological neural networks, especially for hard cases where a 3x time speed up can be achieved. Presentation is clear and the work should be relatively easy to reproduce.

**Weaknesses:**

I find a 3x speed up only for really difficult cases to be of modest improvement. Given currently powerful computers, training a biological style small recurrent neural network is relatively fast which reduces the practical significance of this work.

**Questions:**

Does the training procedure affect the type of synaptic connectivity solution learned? This should be a interesting direction to explore.

---

> ### Author Rebuttal · Authors · 2023-08-10
>
> > This paper proposes a way to speed up biological RNN training by up to 3x using well placed skip connections during training time. This paper will be of interest to researchers training biological neural networks, especially for hard cases where a 3x time speed up can be achieved. Presentation is clear and the work should be relatively easy to reproduce.
>
> We thank the author for the review and helpful comments.
>
> > I find a 3x speed up only for really difficult cases to be of modest improvement. Given currently powerful computers, training a biological style small recurrent neural network is relatively fast which reduces the practical significance of this work.
>
> We understand the concern of the reviewer. Due to the order in which we presented our results, it is possible that too much emphasis may have been unintentionally diverted to the 3x speed up result. While it is one of our significant claims, we have actually presented an even stronger result in our paper that we would like to divert the reviewer’s attention towards. From our abstract, our main claims are:
>
> (1) up to 3x training speedup for biological RNNs on a range of standard cognitive tasks
> (2) enable convergence on cognitive tasks that prove challenging to train with conventional methods
>
> To better present our results, we would like to reorder these two claims to better emphasize the (originally) second claim:
>
> (1) our techniques enable RNNs to successfully learn tasks that are otherwise impossible with conventional methods (rephrase of point 2)
> (2) on simpler tasks where direct training is still possible, we achieve up to a 3.1x reduction in training iterations and 2.7x reduction in training time (rephrase of point 1 with an additional result involving wall-clock training time)
>
> These results are backed by Table 1 in the main text (training iterations) and Table R1 in the uploaded rebuttal PDF (wall-clock training time). In addition, we have reviewed, in Appendix A, a full range of more complex tasks requiring long-term dependencies in neuroscience that our method can potentially be applied to, which again are impossible to train directly without the use of artificial gating elements.
>
> > Does the training procedure affect the type of synaptic connectivity solution learned? This should be a interesting direction to explore.
>
> We thank the reviewer for the helpful suggestion. We will discuss this as future work in a future revision if given the opportunity.
>
> Overall, we hope that our response will improve the reviewer’s impression regarding the significance and scope of our work.

---

> > ### Comment · Reviewer_CT76 · 2023-08-10
> >
> > For the harder tasks, have you tried to train for longer and show it's still impossible to train. Is there a trend towards convergence?

---

> > > ### Author Response · Authors · 2023-08-10
> > >
> > > Yes, the training converges to some low but constant performance, which is a well-known signature of vanishing gradient problems. Skip connections allowed the gradient landscape to be more stable and ultimately resulted in the successful learning of the harder tasks.
> > >
> > > We thank the reviewer for the quick reply and consideration. We will add more training details regarding this in our final revised version if given the chance.

---

### Author Rebuttal · Authors · 2023-08-10

We thank all reviewers for their time and effort in reviewing our submission. We will address common issues here.

Several reviewers have suggested moving information between the main text and supplementary. After also taking into account additional work requested by reviewers, we have arrived at the following outline:

**Current outline**
Main text
1. RNNs as models of the brain
2. Gradient stability issues
3. RNN architecture
4. Review of model parameters
5. Difficulty in learning long-term dependencies
6. Introduction to CD, SCTT, DASC
7. Network stability
8. Network dynamics
9. Training steps required on cognitive tasks
10. Hyperparameter exploration
11. Strategic skip connections
12. Discussion

Supplementary
1. Long-term dependency tasks in neuroscience
2. Review of skip connections
3. Exact model setup and task descriptions
4. Full training results
5. Rule reversal task

**Updated outline**
Main text
1. RNNs as models of the brain
2. Gradient stability issues
3. Brief review of gating mechanisms and other solutions (new)
4. Brief review of skip connections (previously in supplementary)
5. RNN architecture and model setup (previously in supplementary)
6. Difficulty in learning long-term dependencies
7. Introduction to CD, SCTT, DASC and their dynamics (previously two separate sections)
8. Network stability (proof in supplementary)
9. Training steps and wall-clock time results (new)
10. Stating results from hyperparameter exploration
11. Rule reversal task with strategic skip connections (previously in supplementary)
12. Discussion

Supplementary
1. Long-term dependency tasks in neuroscience
2. Extended review on biological plausibility of gating mechanisms (new)
3. Review of other existing methods (new)
4. Review of model parameters (previously in main text)
5. Task descriptions
6. Mathematical proof on stability results (new)
7. Full training results with hyperparameter exploration (previously in main text)

**Novelty**
Reviewer T2Cs had expressed concerns about the novelty of this work. In short, we claim that SCTT confined to training only (but not regular SCTT) is an original contribution from this work. All aspects of DASC are also completely novel. All analyses and experiments that follow are also original to this work.

**Wall-clock training time**
Reviewer nbN6 has raised issues about the validity of using the number of training iterations required as a metric for training speed. To further reinforce our results, we have provided the wall-clock training time and total floating point operations performed during training. These results can be found in Table R1 in the rebuttal PDF (note that only tasks requiring long-term dependencies are shown due to space constraints). Combined with the results from Table 1 of the main text, we make the following conclusions:

On two tasks (delayed decision making and multi-sensory delayed decision making), DASC is able to achieve a 2.0x and 2.7x speed up in terms of wall-clock training time respectively compared to direct training.
On the same tasks, DASC requires 2.3x and 3.1x less training iterations to successfully train the networks.
On two other tasks (delayed go and delayed anti-response), direct training by backpropagation was unsuccessful, but DASC (and SCTT) succeeded.

These results suggest that our proposed methods are able to train RNNs to perform tasks that are otherwise impossible by direct training, and also achieve up to 2.7x speed up in wall-clock time and 3.1x reduction in training iterations on simpler tasks. On the same note, we would also like to address the performance of our methods on the rest of the tasks. These tasks are easily trained by direct backpropagation through time (BPTT) and widely appear in neuroscience literature. Our focus of this work is on tasks where direct BPTT is challenging – these tasks have been extensively reviewed in Appendix A and represent future work.

**Clarification to Figure 3B-C**
Several reviewers have expressed confusion regarding Figure 3B-C. We first clarify the hyperparameters of each of the three methods, shown in Figure R1 of the rebuttal PDF.

Coarsened discretization (CD): initial skip length and step count
Skip connection through time (SCTT): skip length, initial skip ratio and step count
Dynamics-aligned skip connection (DASC): skip length, initial skip ratio and step count

The purpose of Figure 3 is to understand how RNNs trained with our methods behave when simulated normally without any modifications to the dynamics. The exact steps required to reproduce Figure 3B-C are shown in Figure R2 in the rebuttal PDF. We first train RNNs for a small number of iterations using either of the three methods. Throughout training, the skip ratio and skip length remain unchanged. The step count is therefore 0 by definition. We then compare the neural activity when simulating the network with and without the method and compute their mean squared error. We do the same for task performance and compute the absolute difference in performance. These metrics represent the degree in which the modified dynamics differ from the true dynamics. The time constant is 20x the base time discretization. We have found empirically that this is approximately the skip limit for CD to remain stable. The data points for CD therefore do not extend into the dark region of the plots in Figure 3B-C.

**Clarification to Figure 4B**
Reviewer nbN6 has also expressed confusion about the axes of Figure 4B. We have clarified them by using clear color codes in Figure R3 of the rebuttal PDF with reference to Figure R1. The x-axes represent the initial skip ratios of SCTT and DASC, as well as the step counts of all three methods. As requested, we have also split Figure 4B into two rows, where the bottom (top) row represents results from training on tasks that do (not) require long-term dependencies.

We hope our responses to all reviewers help provide a clearer picture on the significance of our work.

---

### Decision · Program_Chairs · 2023-09-21

**Decision:**

Accept (poster)

**Comment:**

Most reviewers are in favor of acceptance due to the clear exposition of the effect of discretizing continuous-time RNNs and the development of their DASC method for combining benefits of SCTT and CD as well as the demonstration of the usefulness of gradually reducing these components over training to allow a test-time continuous RNN with appropriate dynamics.

Two reviewers mentioned that a comparison to LSTMs and GRUs would be necessary/nice to have.   Reviewers are encouraged to add this in the final version.